# Effects on Early Monsoon Rainfall in West Africa due to Recent Deforestation in a Convection-permitting Ensemble

Julia Crook[1], Cornelia Klein[2,3], Sonja Folwell[2], Christopher M. Taylor[2,4], Douglas J. Parker[1,5,6], Adama Bamba[7], Kouakou Kouadio[7,8]

[1] School of Earth and Environment, University of Leeds, UK
[2] UK Centre for Ecology and Hydrology, Wallingford, UK
[3] Department of Atmospheric and Cryospheric Sciences, University of Innsbruck, Austria
[4] National Centre for Earth Observation, Wallingford, UK
[5] National Centre for Atmospheric Science, University of Leeds, UK
[6] NORCE Norwegian Research Centre AS
[7] Laboratory of Material Sciences, Environment and Solar Energy (LASMES), University Felix Houphouet Boigny (UFHB), Abidjan, Cote d'Ivoire
[8] Geophysical Station of Lamto, BP 31, N'Douci, Cote d'Ivoire

*Correspondence to*: Julia A. Crook (j.a.crook@leeds.ac.uk)

**Abstract.** Tropical deforestation can have a significant effect on climate, but research attention has been directed mostly on Amazonian deforestation. The southern part of West Africa (a region dependent on rain-fed agriculture and vulnerable to droughts and flooding) has seen significant deforestation since the 1950s. Many previous tropical deforestation studies have used idealized and exaggerated deforestation scenarios and parameterized convection models. In this study we estimate for the first time realistic historical deforestation from the Land Use Harmonization dataset in West Africa and simulate the impacts in a 5-day ensemble forecast in June using a convection-permitting regional climate model. We find that sensible heat flux increases at the expense of latent heat flux in most deforested regions and rainfall increases by an average of 8.4% over deforested pixels from 18:00-6:00 UTC, whereas changes are much less pronounced during the day. Over large areas of deforestation approx. 300 km inland (e.g., West Guinea) the roughness-length- and thermally enhanced convergence during the afternoon and evening occurs over the deforested areas resulting in increases in rainfall with little impact from reduced daytime humidity. In areas of coastal deforestation (e.g., Cote d'Ivoire), increased winds drive the sea breeze convection inland, resulting in evening rainfall reductions over the deforested area but increases further inland, in line with observations. We suggest our results would not be replicated in parameterized convection models, which are known to struggle with capturing peak convective activity in the late afternoon and long-lived nocturnal rainfall, and with reproducing observed surface-rainfall feedbacks.

## 1 Introduction

The impact of tropical deforestation on climate has been a research area since the early 1990s but there has been particular emphasis on studying Amazonian deforestation due to the Amazon's large size (Lawrence and Vandecar, 2014; Spracklen et al., 2018). In contrast, there has been much less emphasis on West African deforestation. The impact of deforestation on rainfall is dependent on the spatial scale of land cover change, the surface flux characteristics of the replacement land cover, the nature of the rain-bearing systems, and the potential role of topographic and sea-breeze circulations, which makes it difficult to extrapolate what would happen in one region from what happens in another (Lawrence and Vandecar, 2014). There has been considerable deforestation during the 20th Century in the southern part of West Africa (Aleman et al., 2018), with pockets of deforestation driven by the rapidly growing population (Brandt et al., 2017). West Africa has experienced prolonged drought since the late 1960s, but more recently rainfall has somewhat recovered. It is likely that large scale patterns of sea surface temperatures have caused these droughts, while it has been suggested that land use change may have also had some impact (Wang et al., 2004).

Although there is consensus that tropical deforestation causes local (near surface) warming, the impact on rainfall is much more dependent on the spatial scale, extent, and location of the deforestation. Impacts on temperature have been determined from models and observations (Duveiller et al., 2018; Alkama & Cescatti, 2016; Perugini et al., 2017). Deforestation causes increases in albedo which cause cooling, but across the tropics this effect is more than offset by a shift in surface fluxes from latent to sensible heat, resulting in net warming of the atmosphere. The shift from latent to sensible heat flux is due to the lower leaf area index and shallower rooting; the reduction in surface roughness reduces turbulent exchange of heat from the surface, also contributing to a warmer land surface (Spracklen et al., 2018). These modifications to energy fluxes can affect rainfall but changes to rainfall due to deforestation have largely only been studied in models (Perugini, et al. 2017). Studies of continental scale deforestation using global climate models have shown reductions in rainfall over the deforested areas, especially during drier seasons (e.g., for complete deforestation of tropical African rainforests see Werth and Avissar, 2005; Semazzi and Song, 2007). However, when more realistic deforestation scenarios were applied to the Amazon, reductions in rainfall were found to be less, and patterns of rainfall shifted, with some regions having increased rainfall (Medvigy et al., 2011). Abiodun et al. (2008) investigated two idealized but reasonably large-scale deforestation scenarios over West Africa and showed that both could have a significant impact outside the deforested area as well as locally, enhancing the monsoon flow either by changes in meridional temperature gradient (desertification scenario) or by surface roughness changes (deforestation scenario). Overall, they found a reduction in rainfall due to enhanced moisture transport out of the area. Boone et al. (2016) analyzed the impact of estimated land use/land cover change since the 1950s in West Africa in several models. Their somewhat idealized prescribed land cover change represents a worst-case degradation scenario, and they find a reduction in rainfall across the Sahel in all models and a shift in rainfall to the south in some models. Satellite observations of Rondonian (Amazonia) rainfall over deforested areas suggest an increasing rainfall trend in the dry season over the 20th century, but a decreasing trend in the wet season (Chagnon and Bras, 2005). A new observational study for Southern West Africa (Taylor et al, 2022) has identified that

mesoscale deforestation locally enhances the frequency of daytime convective activity, an effect which is particularly
pronounced where coastal deforestation enhances storms triggered by sea-breezes. A positive correlation between forest minus
non-forest differences in sensible heat (from flux tower measurements) and cloud cover (from satellite observations) has been
found globally (Xu et al., 2022), with tropical regions typically showing increased sensible heat and cloud due to the small-
scale deforestation.

Differences in surface heat fluxes between clearing and adjacent forest induce mesoscale circulations similar to land-sea
breezes, bringing moisture rich air from over the forest to over the clearing and, if these circulations are strong enough, result
in enhanced rainfall (Souza et al., 2000; Garcia-Carreras et al., 2011; Hartley et al., 2016). Feedbacks on the atmosphere from
land surface heterogeneity have been shown to be improved for high resolution convection-permitting models (CPM)
compared to parameterized convection models (PM) (Taylor et al., 2013) and therefore high-resolution PM may fail to produce
the observed enhanced convective rainfall at vegetation boundaries. Many of the deforestation-scenario modelling studies have
used low resolution global climate models. However, mesoscale models can reproduce the deforestation-induced mesoscale
circulations which have also been observed in Amazonia (D'Almeida et al, 2007). Khanna et al. (2017) used an 8km scale
model to simulate Rondonian rainfall over deforested areas using 1980s and 2006 land cover. Although their 8km model can
explicitly represent some of the larger scale convection their cumulus parameterization was used for smaller scale convection.
Consequently, they looked at the top of boundary layer humidity as well as rainfall changes. They found that the enhanced
rainfall/humidity over the deforested area in the 1980s was due to thermally driven enhanced mesoscale circulations, whereas
the more recent larger scale of deforestation caused enhanced rainfall/humidity only on the downwind side with roughness
length changes inducing dynamically driven mesoscale circulations. CPM simulations should provide added value to
representing the climate impacts of land use change (Vanden Broucke et al., 2017), because they better represent the differences
in diurnal cycles of surface fluxes over different vegetation types. This is partly due to the higher resolution but largely due to
representing the diurnal cycle of convection better than PM.

In this study we evaluate the changes in rainfall in a 5-day ensemble due to realistic historical deforestation (since 1950) in
West Africa using a CPM and analyse the causes of the changes. We use a 5-day ensemble rather than a longer-term simulation
so that we can look at how local physical processes respond to deforestation in a statistical way without diverging synoptic
conditions. To our knowledge, no other studies have used a CPM or such a methodical way to determine the historical
deforestation scenario to estimate effects of tropical deforestation on rainfall in this region. We analyze the diurnal cycle of
various diagnostics to understand the causes of rainfall changes. We look in detail at two subregions, one where the rainfall
changes are primarily thermally driven and one where the rainfall changes are primarily dynamically driven. The model and
simulations are described in Section 2, the effect of deforestation on the different diagnostics are presented in Section 3, a
comparison with other West African studies is presented in Section 4, and conclusions are presented in Section 5.

## 2 Model Setup and Methods

### 2.1 Modelling Strategy

We use the Met Office Unified Model (UM v8.2) with atmosphere and land surface components, run at 4 km resolution over West Africa (20° W–20° E, 0–25° N), as described in Crook et al. (2019). The model includes a convection parametrization (Gregory & Rowntree, 1990) with closure based on the convective available potential energy. However, this parameterization is severely restricted by adjusting the relaxation time, and a sub-grid Smagorinsky-type turbulent mixing scheme is employed, allowing explicit convection. While a 4km model is at the edge between the grey zone and truly convection-permitting resolution (Prein et al 2015), this model has been shown to represent the diurnal cycle, the intermittency of convective rainfall, the propagation of convection, the location and the lifetimes of deep convective storms in West Africa more accurately than the equivalent 12km parameterized model when compared to CMORPH rainfall, TRMM radar (2A25) and SEVIRI brightness temperature (Crook et al., 2019, simulation V_CP4 therein). It does, however, have storms that are often too intense and never reach the size of the largest observed storms, and the small storms produce too much of the total rainfall. It has also been shown to capture the observed relationships between surface flux patterns and convective triggering, unlike the 12km parameterized model (Taylor et al 2013). Although our simulations are not set up as a forecast and we would not expect a perfect match to observations, we find that the ensemble mean rainfall over the 1$^{st}$-5$^{th}$ June 2014 period is, like CMORPH, mostly confined to south of 12° N and with the most intense rain near the south coast. However, the ensemble mean has a wet bias across the domain around 10° N and over the Cameroon Mountains, and a dry bias near the coast in Cote d'Ivoire and Ghana and these biases are significant compared to the 1$^{st}$-5$^{th}$ June standard deviations found from nine years of CMORPH (supplementary Fig. S1). Despite this, systematic regional biases that affect both our forested and deforested simulations equally will not affect the rainfall change signal linked to deforestation that we are interested in. Given this is a process study, it is the model skill in correctly capturing rainfall timing within the diurnal cycle and in representing the characteristics of convective storms (as demonstrated in Crook et al., 2019) that is most important for this work. The realistic representation in timing, storm lifetimes and storm precipitation intensities provides confidence in our model results on convection responses when surface roughness and flux patterns change locally due to deforestation.

To assess the impact of recent West African deforestation on rainfall, we produced two 10-member 5-day ensembles, the first using current land cover and the second using an estimate of 1950's land cover. Both current and 1950 vegetation ensembles were run at 4 km resolution for 5 days from 1st June 2014 conditions over West Africa. We chose early June because at this time of year there is some rain in the region up to about 15° N, while still being early in the monsoon season when it is expected that sensitivity to deforestation is high, i.e., soils are still relatively dry, and the evaporative advantage of forests compared to shallow-rooted vegetation is expected to be high. Although earlier in the year the soil would have been drier near the coast, there were fewer rain events making detection of impacts on rainfall challenging. The ensemble approach with 10 paired members allows us to evaluate the uncertainty in the modelled response to forest cover change that is linked to internal variability. At the same time, by simulating only 5-day time slices in "forecast mode", temporal divergence of the synoptic

conditions between the ensemble members is minimized. Differences between the two ensembles can therefore be attributed to the imposed deforestation in the absence of large-scale circulation feedbacks.

Sea surface temperatures and boundary conditions were prescribed from ERA-Interim data (Dee et al., 2011) every 6 hours. The ensembles were generated by starting each ensemble member from the previous current land cover ensemble member dump file at the end of the first day with the time reset to 1st June 00:00 but resetting the soil moisture to the climatology. Climatologies for soil moisture were produced as in Crook et al. (2019) from 14-year off-line land surface model (JULES) runs using either the current or 1950s vegetation.

JULES (Best et al., 2011) is a modular land surface model, handling exchange of heat, moisture and momentum with the atmosphere, soil moisture hydrology split into four soil layers of thicknesses 0.1, 0.25, 0.65, and 2.0 m, surface and sub-surface runoff parameterizations, and a vegetation model representing five different plant functional types (PFTs): Broadleaf trees (BT), needleleaf trees, C3 grass, C4 grass, and shrub. Plant and soil properties (e.g., albedo, roughness length, leaf area index (LAI), soil conductivity and soil thermal capacity) and fractions of each PFT as well as urban, ice and water (each modelled

as a tile within a grid box) per grid box are specified through ancillary files and model configuration. The amount of moisture in each soil layer available to each PFT depends on soil properties and on the root density of each PFT in each soil layer, whereas water available for evaporation from bare soil only comes from the uppermost soil layer. Due to a lack of suitable in-situ flux measurements in the study region, JULES' translation of deforestation into changes in net radiation and surface flux partitioning cannot be directly validated. However, based on known effects of tropical deforestation in better studied regions

(e.g., Silvério et al., 2015 (Amazon); Peng et al, 2014 (China)), the flux differences between forested and deforested areas in the default JULES configuration did not appear sufficiently realistic, as detailed below. Such unrealistic responses to land cover change in "out-of-the-box" land surface schemes are not unusual (Pitman et al., 2009; Boone et al., 2016) and even in the most recent earth system models there is a large difference in behaviour (Boysen et al. 2020). We therefore implemented the following modifications in JULES to simulate a more plausible depiction of deforestation.

**2.1.1 Changes to leaf area index**

Firstly, we found that the monthly climatology of LAI used in standard JULES/MetUM simulations produces an unrealistic seasonal cycle over Southern West Africa. This problem was traced to the treatment of missing data in the creation of the LAI field whereby cloudy pixels were erroneously assigned LAI of zero. Given that during boreal summer, cloud cover is extensive, the effect was to introduce a marked minimum in LAI across the region. Instead, we used an alternative LAI field (Semeena

et al., 2021) based on the Global LAnd Surface Satellite (GLASS) LAI product (Xiao et al., 2016) which exhibits a broad maximum in LAI during these months (see supporting material Fig S2).

**2.1.2 Changes to root water extraction for broad leaf trees**

The JULES model expresses the impact of soil water stress on transpiration via a dimensionless stress factor (FSMC), based on soil moisture relative to texture-dependent critical ($\theta c$) and wilting ($\theta w$) points (Best et al., 2011, see equation 52). A value

of 1 means there is no water stress at all and a value of 0 means the plant is totally water stressed. For each plant functional type, this is computed by weighting FSMC at each soil layer by its assumed vertical root profile, where the root density in each layer follows an exponential distribution with depth as given by equation 50 in Best et al. (2011).

For the BT type of key interest in our study, the root weighting for the lowest soil layer (1-3 m) is substantial, corresponding to 0.55. We found in offline simulations that throughout the first few months of the rainy season, this lowest soil level remained strongly water-stressed, suppressing the root-weighted FSMC (and hence transpiration) from trees. By contrast, shallower-rooted grasses exploit the wetter soil closer to the surface and produce much larger transpiration than the trees. We consider that in reality, trees are well adapted in terms of where they take their water from, and we hence modified JULES for BT so that FSMC is taken from the maximum value of the 4 levels, rather than its root-weighted mean.

### 2.1.3 Under-canopy evaporation switched off

In JULES, direct moisture evaporation from the upper soil layer occurs primarily from the soil tile. However, for the vegetated tiles, total evapotranspiration is made up of transpiration from the leaves plus a contribution from soil evaporation based on the fraction of bare soil visible through the vegetation canopy. Under some circumstances, this bare soil contribution in vegetated tiles is known to be too large (Van Den Hoof et al., 2013). We found that this source could (counter-intuitively) enhance evapotranspiration following deforestation and we therefore switched the bare soil contribution off for all plant tiles.

### 2.2 Creation of the deforestation scenario

The key aim of this study is to explore atmospheric and rainfall responses to a deforestation scenario that plausibly captures historical forest change. This crucially requires a realistic and consistent estimate of past and current forest cover, necessitating an informed merging of diverse land cover and land use information sources for use in JULES. The major steps for this novel approach are described in the following. The current land cover is based on the European Space Agency's Land Cover Climate Change Initiative (CCI) land cover dataset, version 1.4 for the 2008-2012 epoch (Poulter et al., 2015), which classifies each 300 m pixel as one of 23 United Nations Land Cover Classification System (UNLCCS) classes. These are mapped to the model PFTs according to Poulter et al. (2015). Creating a plausible map of vegetation in 1950 consistent with the JULES model is more challenging, and we summarize the process in Fig 1a. Firstly, we use estimates of land use change from 1950 to the present from the Land Use Harmonization (LUHv2) dataset developed for use by Earth System Models (Hurtt et al., 2011). This describes the landscape in terms of fractions of different land use types, rather than the PFTs which JULES requires. We therefore developed a mapping procedure specific to West Africa to translate land use types into fractional coverages of PFTs plus the non-vegetation functional types of inland water and bare soil fractions. For this we identified areas of primary land from the World Database of protected areas (UNEP-WCMC and IUCN, 2014). Considering only the nature reserves with the highest level of conservation (IUCN categories I to VI), we extracted the 300m CCI pixels within each protected area. Recognizing that even within these zones, there will be some degree of human disturbance, we identified the UNLCCS class with the highest fractions of BT and Shrub within each protected area. Mean annual rainfall across West Africa provides a

dominant control on the composition of natural vegetation. Using the JULES composition of natural vegetation, as sampled within protected areas, we devised a simple threshold-based mapping of natural vegetation as a function of mean annual rainfall. For rainfall, we used version 2 of the CHIRPS dataset (Funk et al., 2015), downscaled to the 4 km model grid for

195    years 1981 to 2015. The rainfall thresholds and dominant natural land cover classes (Shrub land to bare soil, Shrub land, broadleaved deciduous and broadleaved) and how they relate to model PFTs are summarized in Fig 1b. To create a map of primary vegetation cover in terms of JULES PFTs, we applied the relationships in Fig 1b to the mean annual rainfall map.

In step 2 we apply the change in primary land to produce a map of 1950 land cover, selecting all 4km model pixels within a 0.5° grid box in turn. To avoid introducing artificial 0.5° structure into the high-resolution land cover map a Gaussian-smoother

is first applied to the LUHv2 map of primary land cover change. For each 4km pixel we compare ratios of existing land and primary vegetation to identify the proportion of current primary land and then apply the change to that 4km grid box fraction. In cases where the 4km values are already at or close to the primary land fractions we iteratively apply changes to the remaining pixels so that overall, the 0.5° change is applied. For the remaining non-primary land fraction, we adjust the remaining proportions of C3, C4, shrub and bare soil accordingly. Grid box urban and water fractions are kept at their 2010 values, e.g.,

no account taken of the construction of the Akosombo dam of the Volta River in 1965 or changes in urbanization.

(a)

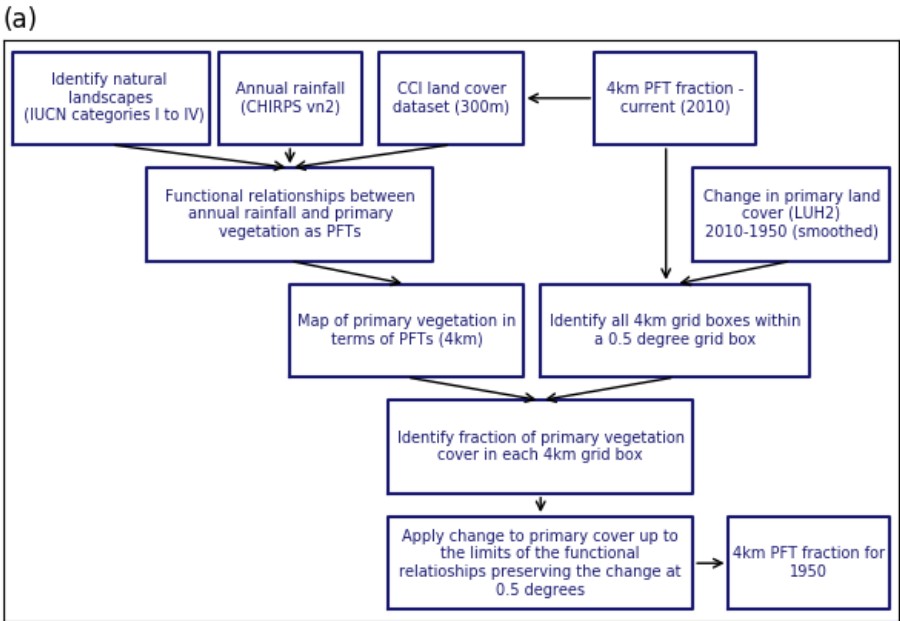

(b)

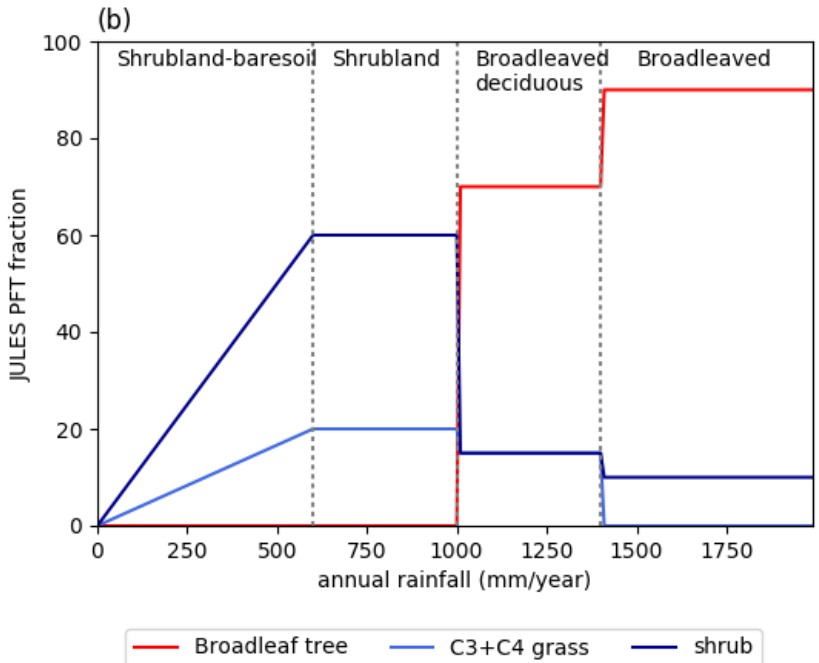

**Figure 1: (a) Schematic showing datasets and processing steps to produce the 1950 PFTs fractions. (b) Rainfall (CHIRPS) thresholds and the dominant "natural" UNLCCS classes identified in the CCI land cover dataset and corresponding PFT fractions identified for the West African domain.**

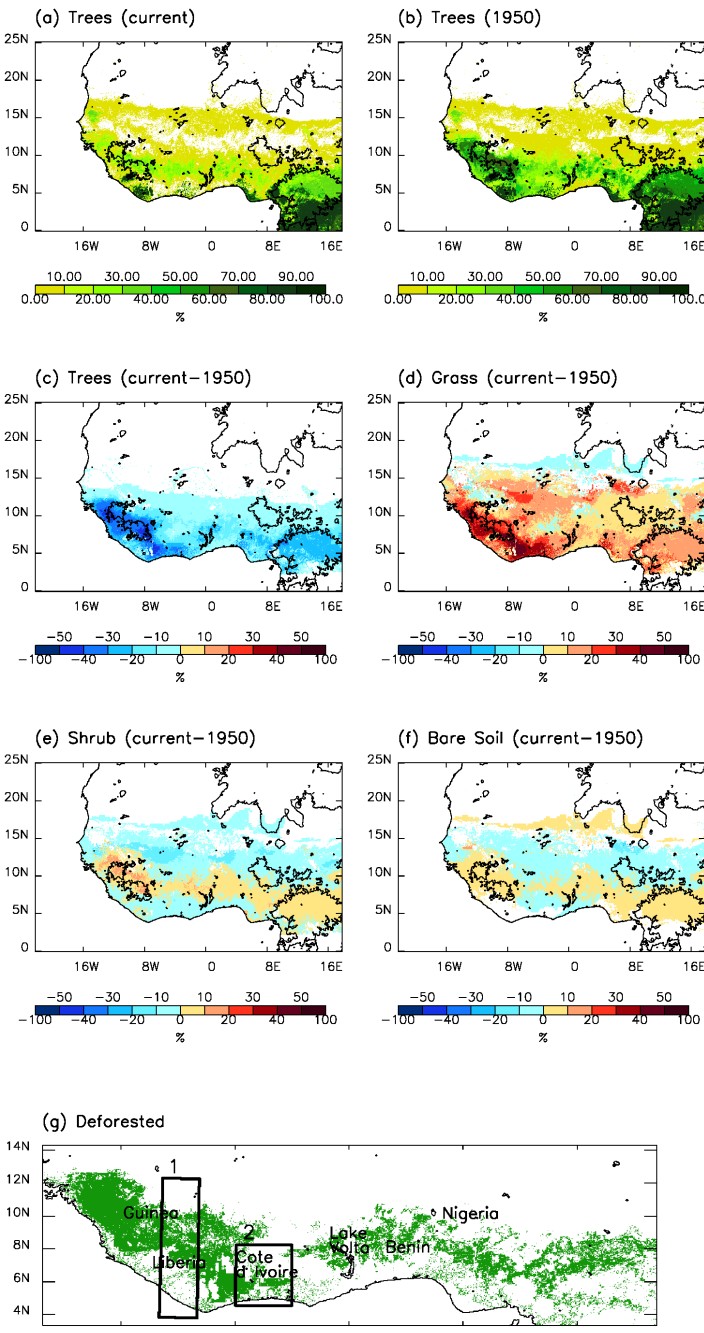

**Figure 2: Tree cover for a) current and b) 1950s, differences in cover of (c) trees, (d) grass, (e) shrubs and (f) bare soil (land higher than 500m shown by black contour) and (g) deforestation mask with 2 sub-regions analyzed highlighted with black rectangles. Deforestation is defined as where 1950 tree cover > 30 %, current tree cover < 30% and change in tree cover > 10%.**

Figure 2 shows maps of the changes in trees, grass, shrubs, and bare soil from 1950 to current in the target region. Trees have largely been replaced with grass although there have been some changes to shrubs and bare soil. We define deforestation as where 1950 tree cover > 30 %, current tree cover < 30% and change (current-1950) in tree cover > 10% (Fig 2g). We use 30% tree cover as separating forest from non-forest as there appears to be a changeover in behaviour of turbulent fluxes around this level (supporting material Fig S3). This threshold was also used by Hartley et al. (2016).

## 2.3 Statistical Significance Tests

Paired student T-tests of the 5-day means of the current and 1950's vegetation ensemble members are used to determine significant changes at the pixel level. In the case of rainfall, the natural logarithm of rainfall is taken first before performing the T-test because rainfall is not normally distributed. We determine significance of changes at the 95% confidence interval ($p<0.05$) for all variables except rainfall, where we relax the significance to the 90% confidence interval ($p<0.1$) due to the patchy nature of rainfall over the short duration of our simulations. When showing field significance over the region we need to consider the fact that some false rejections of the null hypothesis will be made and this increases with the number of individual tests. To overcome this, we use the method of Wilks (2016) to control the false discovery rate (FDR). We use a Benjamini-Hochberg correction (Wilks, 2016 Eq. 3) with $\alpha_{FDR}=2\alpha$ (i.e., $\alpha_{FDR}= 0.1$ for $\alpha=0.05$) to find a new threshold for rejecting the null hypothesis. However, the fact that we have a large number of pixels due to the high resolution of the data imposes a very strict limit on the p values. Also, we do not expect all variables to have changes everywhere in the region, in which case we would not expect field significance to be applicable. In the case of variables which have very patchy patterns of change, we, therefore, just show the significance at the individual pixel level. We highlight where this is the case.

## 3 Results

We first show the impact of the deforestation on surface characteristics of albedo, roughness length and initial soil moisture. We then compare the 5-day means (current vegetation simulations vs. 1950s vegetation simulations) of turbulent and radiative heat fluxes, 1.5m temperature, conditional instability, 10m winds, and rainfall. Finally, we look at two focus regions (Guinea East 10-8°W and Cote d'Ivoire 6-3°W) that exhibit different 1950s winds and deforestation patterns, and where we detect large changes in rainfall caused by locally different thermal and dynamical responses to deforestation.

## 3.1 Albedo, Roughness length and Initial Soil Moisture

The changes in vegetation result in changes in albedo and roughness length as shown in Fig 3a and 3b, respectively. Albedo mostly increases over the deforested areas by up to 0.02 from around 0.15 (i.e., by up to approx. 12 %), although there are some areas with small decreases coinciding with where shrubs have replaced forest. Roughness length decreases over the deforested area by around 0.5m compared to the 1950s, where roughness length is typically 1-3m over the forested area. In

June, the monsoon has reached the southern part of the region and the soil moisture is high enough that soil moisture control on transpiration is weak (tile-weighted soil moisture stress factor (FSMC) >0.8), whereas further north it is stronger (Fig 3c). Trees are able to transpire throughout the dry season due to their deep roots, depleting soil moisture particularly in the deeper soil layers, whereas grasses cannot reach these layers. In the 14-year offline spin-up of soil moisture, this decreased dry season soil moisture depletion for grass cover produces an increase in initial soil moisture at all four levels under current vegetation

compared to 1950s vegetation over the deforested areas. As a result, FSMC is increased (less water stress) by up to 0.1 in some areas. In areas with greater deforestation, even though soil moisture has increased, the trees were able to access more water than the grass that now replaces them, so deforestation causes FSMC to decrease by -0.14 (Fig 3d). In southern areas with coastal deforestation, changes are very small because it has already been raining in this region for 1-2 months, resulting in a high FSMC (low water stress for trees and grass).

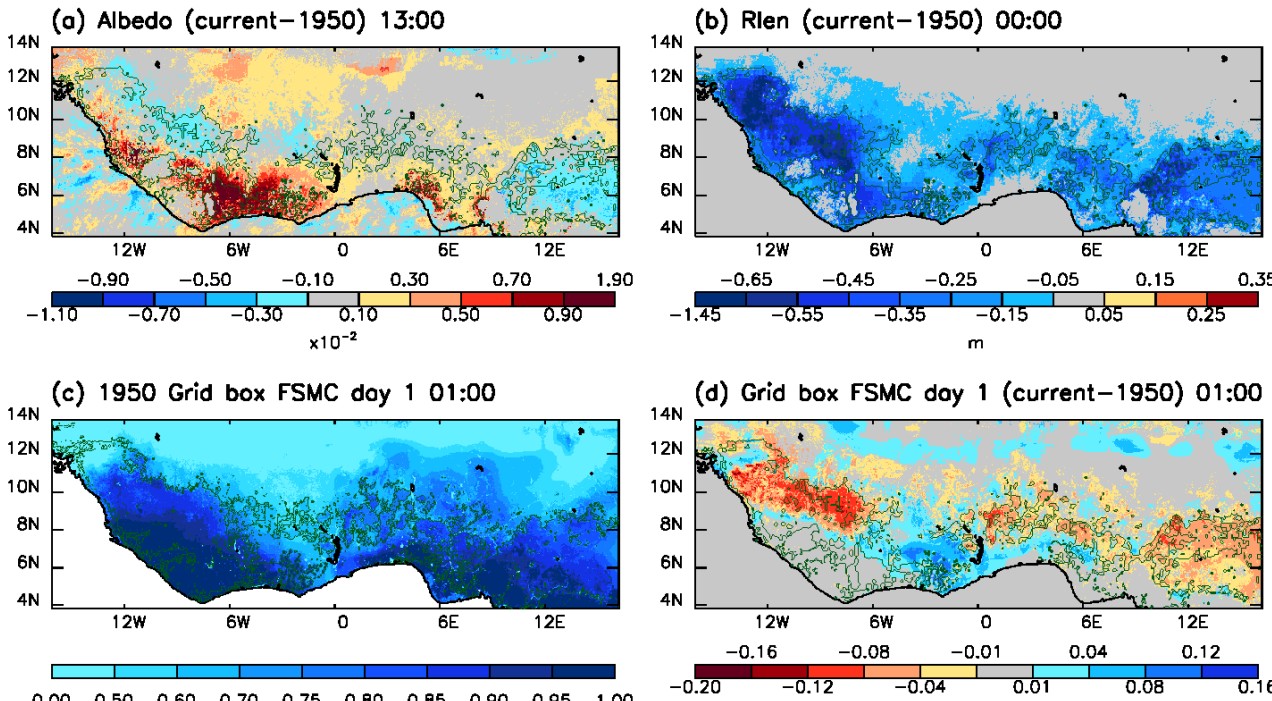

**Figure 3: (a) Changes in albedo, (b) Changes in roughness length, (c) 1950s initial FSMC and (d) Changes in initial FSMC. Green contours show where there has been deforestation in (a), (b) and (d) and the 1950s forested region in (c). Small changes either side of zero are shown in grey.**

**3.2 Turbulent and Radiative Fluxes**

Diurnal cycles of the change in turbulent and net downward surface radiative fluxes over deforested areas are shown in Fig 4a. Peak differences in radiative and turbulent fluxes occur between 11:00-14:00 UTC. Deforested areas show decreased latent heat flux (LH) throughout the day while sensible heat flux (SH), associated with higher surface temperatures, is increased only

until 15:00 UTC. The canopy heat capacity for trees is larger than for grass (areal heat capacities in JULES are 320000, 12000
and 8000 J $K^{-1}$ $m^{-2}$ for BT, C4 and C3 grasses, respectively). This means that during the morning, more heat is absorbed to
warm the tree canopy than grass, whereas later in the day, the heat stored in the tree canopy is released. Therefore, deforestation
causes increases in combined SH+LH in the morning (up to 13:00 UTC) but decreases in the afternoon when also the maximum
in cloudiness and convective activity occurs. In the following, we hence consider spatial patterns of pre-convective conditions
at 13:00 UTC, when daytime heating is strong, radiative flux differences are maximized and perturbations from active
convection are still limited.

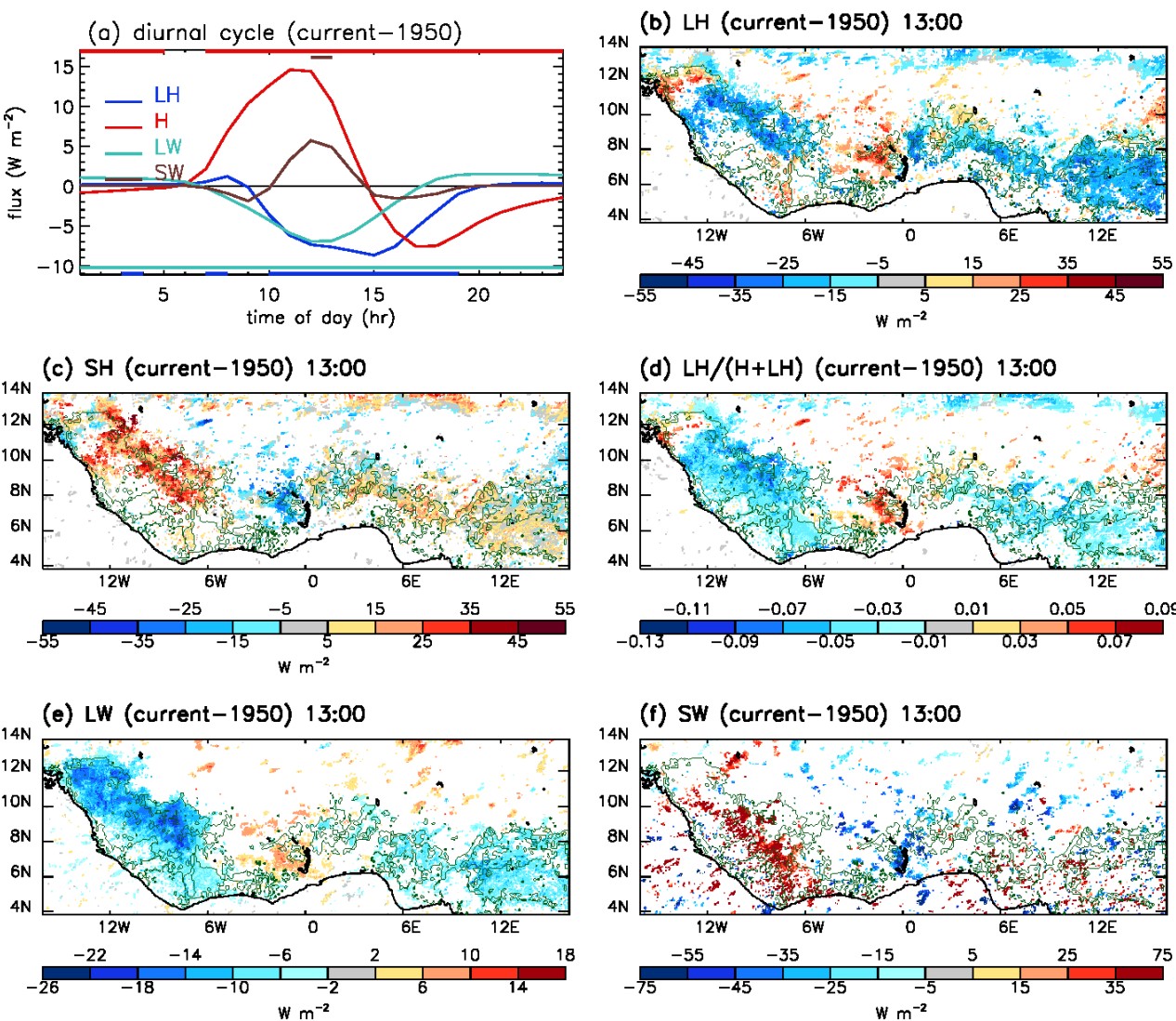

**Figure 4: (a) Diurnal cycle of changes in turbulent and radiative heat fluxes averaged over all deforested pixels with times when
differences are significant indicated by coloured bars at the top and bottom. Differences in fluxes during the day (13:00 UTC): (b)**

latent heat, (c) sensible heat, (d) evaporative fraction, (e) net down surface longwave and (f) net down surface shortwave. **Insignificant changes are shown in white (for SW this is the individual pixel significance, whereas for other variables, FDR correction has been applied), whereas small but significant changes either side of zero are shown in grey. Green contours show where there has been deforestation.**

At 13:00 UTC, deforested regions predominantly have lower LH (Fig 4b) by up to 30 Wm$^{-2}$, higher SH (Fig 4c) by up to approx. 40 Wm$^{-2}$, and lower evaporative fraction (Fig 4d). There are several areas (e.g., around 14° W, 11° N and 3° W, 8° N) where deforestation is accompanied by an increase (rather the expected decrease) in evaporative fraction. This can occur where dry season soil moisture depletion decreased after deforestation, as discussed in Section 3.1.

The day-time net downward longwave radiative flux (LW) decreases by up to approx. 18 Wm$^{-2}$ (i.e., increased emission). Reduced roughness length suppresses turbulent fluxes, (i.e. reduced heat from surface to atmosphere, but particularly LH due to the shift in flux partitioning to greater SH), and therefore increases land surface temperature which increases upward longwave radiative flux (LWu) and decreases LW with no change in downward longwave radiative flux (LWd). Reduced cloud cover decreases LWd enhancing the decrease in LW although this is a smaller effect (Fig 4e). The net downward shortwave radiative flux (SW) is affected by both albedo increases (causes decreases in SW) and by reductions in cloud cover (causes increases in SW) resulting in a patchy overall increase in day-time SW (Fig 4f) by up to approx. 40 Wm$^{-2}$. When averaged over the deforested pixels, the changes in LW outweigh the changes in SW, resulting in a decrease in net radiative flux. However, for individual pixels, SW changes may dominate, and the total radiative flux changes (not shown) are similar in pattern but smaller in magnitude to SW changes.

In summary, averaged across all deforested pixels during the day, the simulations indicate that deforestation increases sensible heat, decreases latent heat and increases long-wave emission from the surface. We do not have multi-site long-term flux observations in this region with which to compare the model, but this behaviour is consistent with pan-tropical analyses of increased air temperature responses to deforestation (Alkama and Cescatti, 2016; Duveiller et al., 2020). However, we note that there are deforested regions within the domain where these flux responses are muted (where 1950s soils are wet), or even reversed. Under both these circumstances, the dynamical (roughness) effect is expected to dominate over a negligible thermal effect, as in the simulations of Khanna et al. (2017).

## 3.3 Near-Surface Temperature and Conditional Instability

Trees shade the surface from the sun during the day, have higher heat capacity and roughness length than grass, and transpire more than grass causing evaporative cooling. Therefore, we expect land to be cooler in forest regions than grass regions and a warming due to deforestation. At night, the enhanced aerodynamic coupling with the atmosphere that forests exert (compared to smoother surfaces), along with the release of heat absorbed by trees in JULES, means that we expect the land to be warmer in forest regions than grass regions, and a cooling due to deforestation. In line with that, Fig 5a shows tropospheric temperature increases during the day up to about 800 hPa and tropospheric temperature decreases during the night up to 900 hPa (with largest temperature decreases confined to below 950 hPa) when averaged over deforested pixels. Figures 5b and 5c show warmer near-surface temperature during the day (up to 1 K) and cooler near-surface temperature at night (approx. -0.5 K) in

the deforested areas, resulting in a net warming over the whole day. Temperature changes are more pronounced where the
310 areal extent of deforestation is larger. Differences also occur outside the deforested areas, e.g., night-time cooling north of the
deforestation in Cote d'Ivoire, likely linked to changes in near-surface winds (discussed later), and daytime cooling to the west
of Lake Volta due to decreased daytime sensible heat in that location (cf. Fig 4c).

There are significant surface pressure differences (up to 0.3 hPa, not shown) that largely follow the near-surface temperature
differences consistent with localized heat-low effects, as discussed by Taylor et al. (2005), but these are more homogeneous
across the deforested areas during the day and patchier at night.

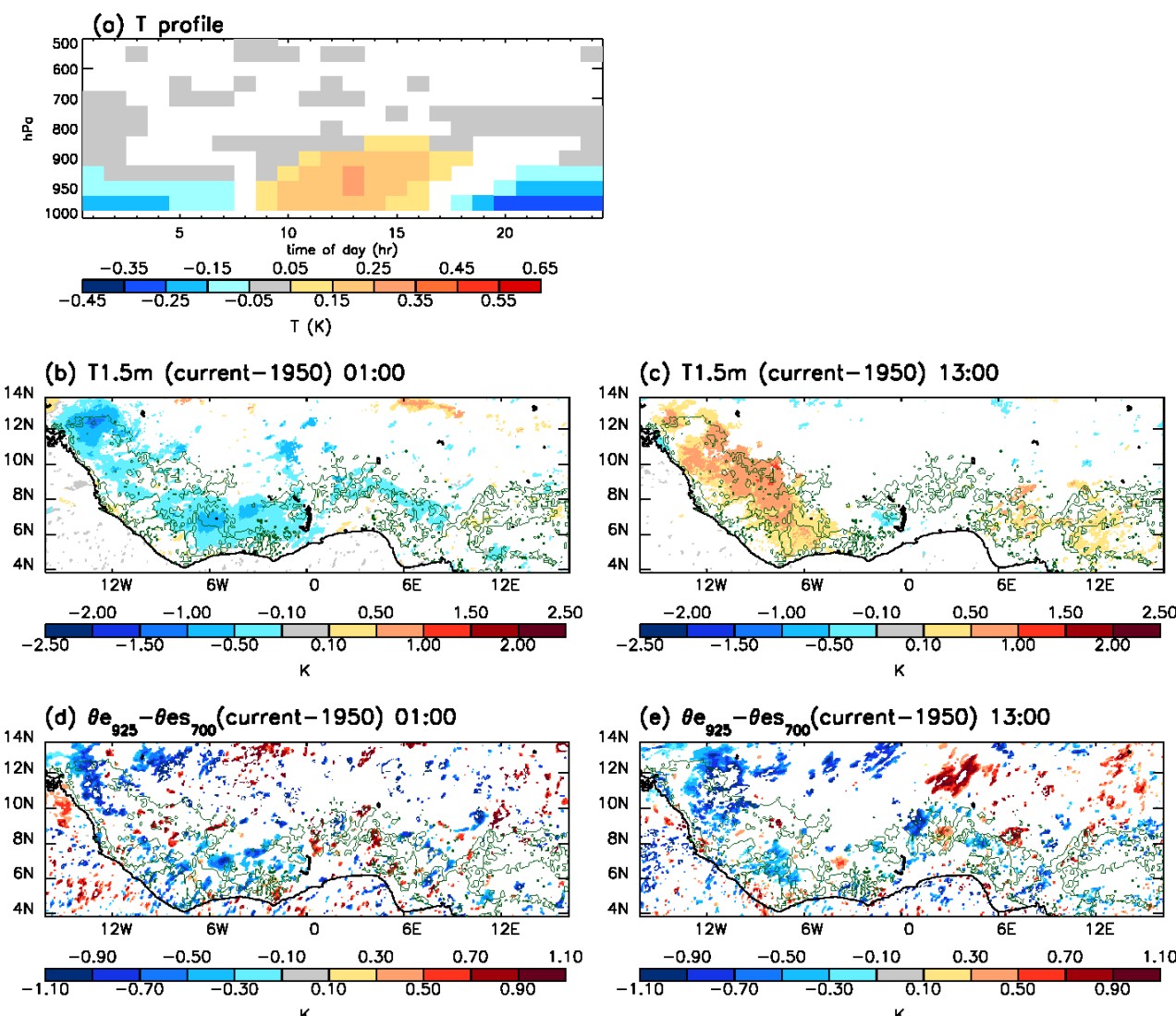

**Figure 5: Changes in diagnostics for (a) diurnal cycle of temperature profile averaged over all deforested pixels, and for night (01:00 UTC) (left) and day (13:00 UTC) (right): (b) and (c) near-surface temperature, and (d) and (e) $\Delta\theta$ defined as $\theta_e$ at 925 hPa minus $\theta_{es}$**

**at 700 hPa. Insignificant changes are shown in white (for Δθ this is the individual pixel significance, whereas for T1.5m, FDR correction has been applied), whereas small but significant changes either side of zero are shown in grey. Green contours show where there has been deforestation.**

The equivalent potential temperature ($\theta_e$) in the boundary layer is a measure of convective available potential energy (CAPE), whereas the saturated equivalent potential temperature ($\theta_{es}$) above the lifting condensation level is a measure of convective

inhibition (CIN). A parcel in the boundary layer is buoyant if $\theta_e > \theta_{es}$. Therefore, the difference between the two gives a measure of conditional instability (Garcia-Carreras et al., 2011). We use $\theta_e$ at 925 hPa minus $\theta_{es}$ at 700 hPa, hereafter called Δθ, as our 2-level conditional instability measure. Figure 5d and 5e show changes in Δθ. Where lower tropospheric temperature increases during the day, which would increase buoyancy, this is typically coincident with humidity decreases, making the atmosphere less buoyant, resulting in overall decreases in Δθ (by up to 1.1K). In fact, Δθ changes are more related to changes in humidity

than changes in temperature (compare Figure 5 (d) and © with supplementary Fig S4). The magnitude of changes in Δθ are greater during the day than the night when temperature and humidity changes are less.

### 3.4 10m Winds

The 1950s 10m winds during night and day are largely south to south-westerly over the region (Fig 6a and 6b). The differences in 10m wind speeds show increases during both night and day over the deforested area (Fig 6c and 6d) as expected due to

roughness length decreasing, and the increases are greatest where the 1950s winds are high. There are only small changes in 10m wind direction (Fig 6e and 6f). The changes in 10m winds depend strongly on location and extend into regions where there have been small amounts of vegetation changes but are not marked as 'deforested' based on our forest change thresholds (see Fig 3b for changes in roughness length). In addition, increases also occur just downwind of deforestation boundaries. Changes in pressure gradients can also modify the winds by increasing the wind into (deforested) low pressure regions and out

of high-pressure regions although this appears to be a weaker effect compared to the roughness length changes. As pressure changes are less pronounced at night (mostly insignificant and less than 0.3 hPa), the changes in winds are largely due to the reduced roughness length (night-time changes in winds strongly correlate with expected roughness length scaling, see supplementary Fig S5). Increases in 10m wind speeds are likely to cause convergence on the downwind side and divergence on the upwind side of deforestation as changes are greater within the deforested area than outside.

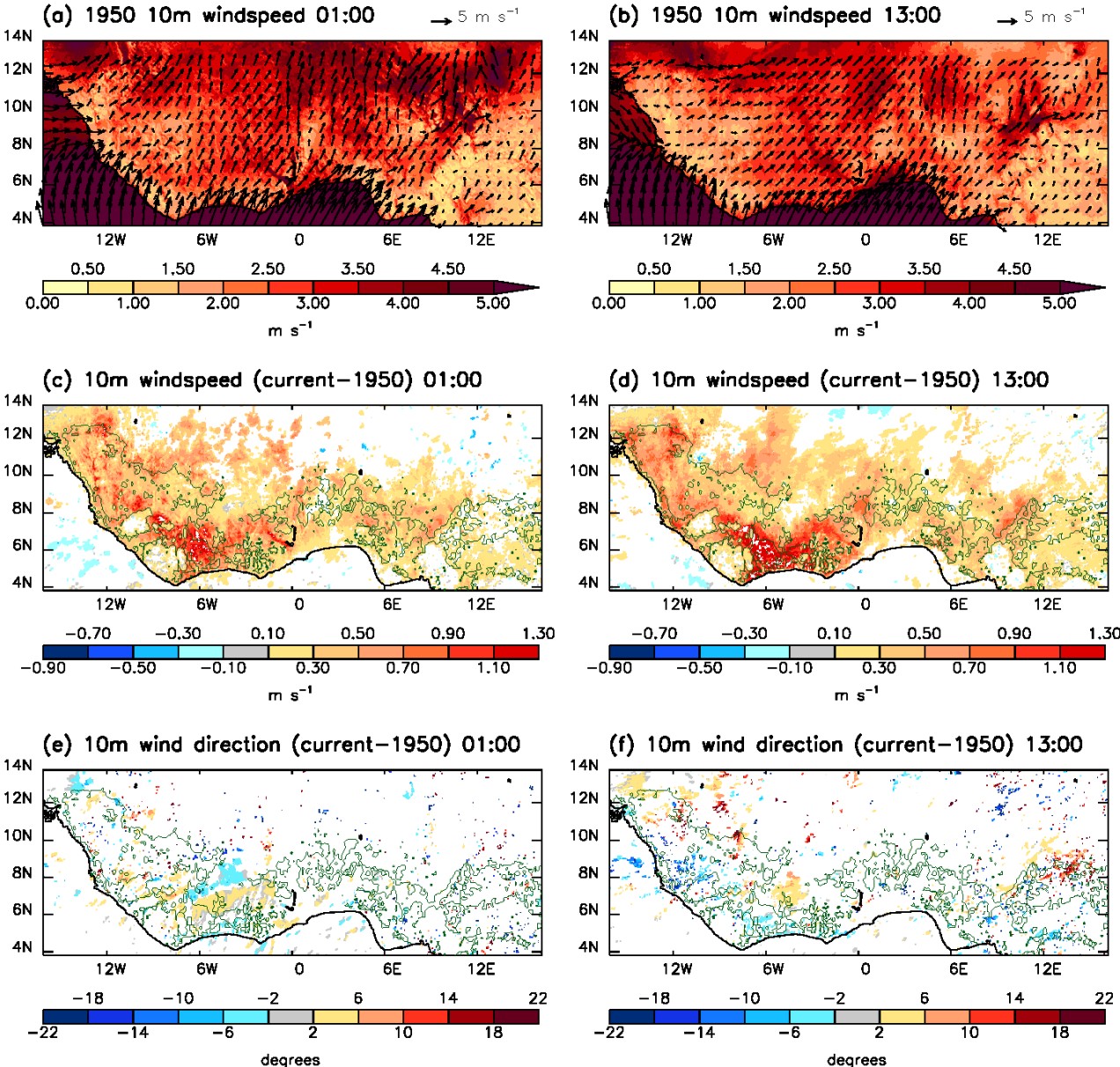

**Figure 6: 10m winds for 1:00 UTC (left) and 13:00 UTC (right). 1950 10m wind (speed in shading, direction show by vectors) at (a) 1:00 UTC and (b) 13:00 UTC. Changes in: (a) and (b) 10m wind speed and (c) and (d) 10m wind direction. Insignificant changes are shown in white (for wind direction this is the individual pixel significance, whereas for windspeed, FDR correction has been applied), whereas small but significant changes either side of zero are shown in grey. Green contours show where there has been deforestation.**

### 3.5 Rainfall

The ensemble-mean daily rainfall accumulation averaged over all land points south of 15° N shows a small but significant increase of +2% (+0.17 mm) in the current vegetation simulations compared to 1950 vegetation simulations (1950=8.35 mm),

and this increases to +6% (+0.64 mm) for deforested pixels (1950=10.18 mm) (Fig 7a and 7b). This behaviour is maximized between the hours of 18:00 and 06:00 UTC with an average of +8.4% and a maximum of +12.8% at 00:00 UTC.

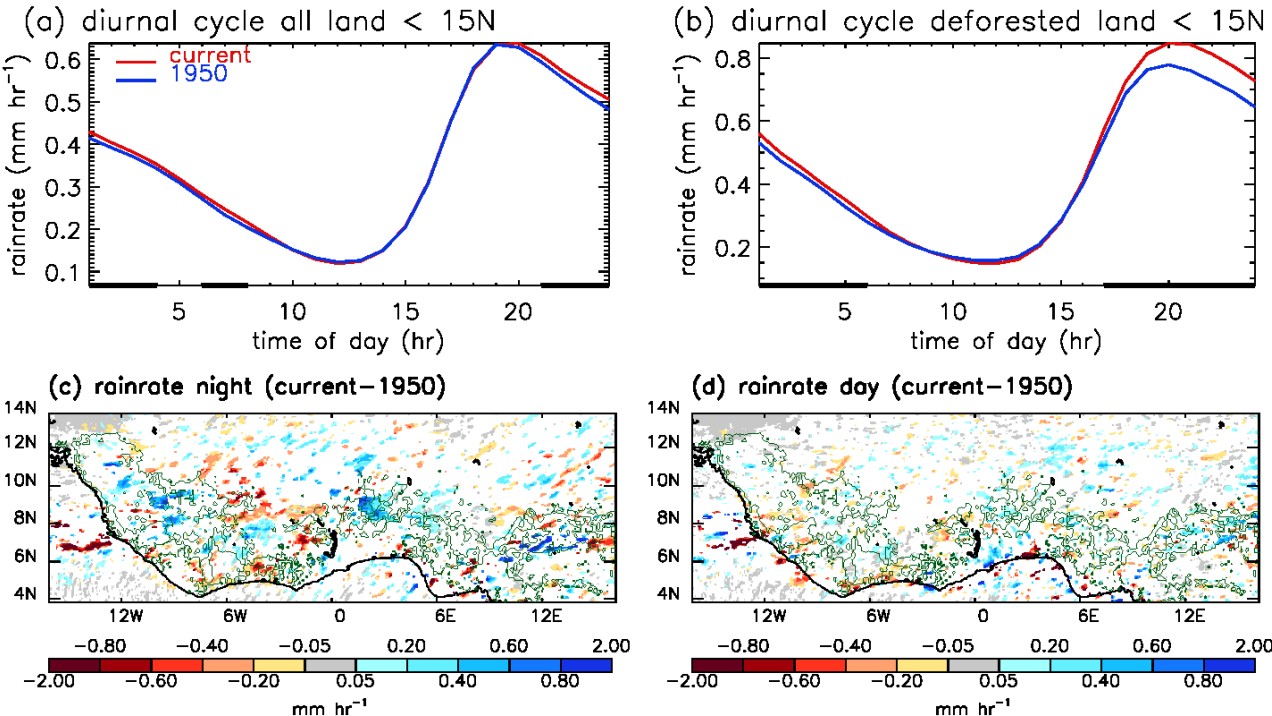

**Figure 7: Diurnal cycle of rainfall with times where differences are significant at p=0.1 indicated by black bar for (a) all land points below 15° N and (b) deforested pixels only. Maps of rainfall differences for (c) 19:00-6:00 UTC and (d) 7:00-18:00 UTC. Insignificant changes at the individual pixels level are shown in white, whereas small but significant changes either side of zero are shown in grey. Green contours show where there has been deforestation.**

Maps of the change in ensemble mean rain rate averaged over the night-time and daytime are presented in Fig 7c and d respectively. They confirm larger and more significant changes at night, with rainfall increases over many deforested areas, though scattered decreases in rainfall and changes in non-deforested regions do occur (e.g., north Côte d'Ivoire). This illustrates that, from this short set of simulations, small-scale rainfall changes are too noisy to draw robust conclusions about the nature of the rainfall response to deforestation locally. Aggregated over all deforested pixels however, an increase in rainfall with deforestation clearly emerges (Fig 7b), consistent with observations for this region (Taylor et al., 2022, hereafter T22). T22 analyzed changes in frequency of convective cores over a 30-year period as a function of LST trends (a proxy for deforestation). They found maximum increases around the late afternoon/early evening convective peak, with enhanced convection persisting downstream for several hours (see their Figure 2) in line with our own results. We used the data of T22 to determine the relative change in core frequency per amount of deforestation and scaled for significant deforestation, thus removing changes unrelated to deforestation. This results in an estimate of 25% increase in relative core frequency, somewhat higher than our 9% relative rainfall change. We would not expect an exact match given the different nature of the data and that the observations span the

entire rainy period in Southern West Africa for a period of 30 years whilst our simulation represents conditions from a single
year in early June.

## 3.6 Drivers of rainfall change in specific regions

Processes governing rainfall changes are dependent on proximity of the deforestation to the coast, location of 1950s rainfall and strength of the sea breeze, the soil wetness and the extent of deforestation (larger areas of deforestation and drier areas have greater temperature differences). To demonstrate this, we now assess in detail the changes in two specific focus regions (shown in Fig 2g), chosen for their contrasting soil wetness, extent of deforestation and proximity to the coast. In the first case (Guinea East, 10-8° W, Fig 2g box 1), the extensive deforestation is up to 400 km from the coast, a region that is 1-2 months into the rainy season for the simulated period. The recent start of rainfall after the dry season means that evapotranspiration is still limited by soil moisture approx. 200 km or more inland (i.e., FSMC<1) such that deforestation induces a decrease in evaporative fraction and atmospheric warming. This decrease in evaporative fraction with deforestation is also true for the Sierra Leone/ Guinea West (around 13W) region. We thus consider our chosen Guinea East box as representative for the deforested latitudinal band across these regions. The second region (Cote d'Ivoire, 6-3°W), with marked deforestation 40-200 km from the south coast, was chosen for its earlier start to the rainy season meaning that soil moisture is not limited and consequently has only a weak control on evaporative fraction during the simulation. Moreover, rainfall in this second case is strongly influenced by the daytime penetration of the sea breeze. Whether the rainfall changes in these regions are predominantly dynamically or thermally driven depends on the local characteristics, which will we investigate in the following. Between 56-70% of rainfall in these sub-Sahelian regions is produced by Mesoscale convective systems (MCSs) (Maranan et al., 2018). These storms tend to initiate in the afternoon to early evening and last for many hours. We use an MCS tracking algorithm on 15-minute rain rate fields to find storms where rain rate exceeds 1 mm/hr and where the storm reaches at least 1000 $km^2$ in area at some point in its lifetime. The algorithm is described in Crook et al. (2019). For both focus regions (Fig.2g boxes), we determine frequency and properties of storms over land at each time of the day in both 1950 simulations and current simulations to understand the changes in rainfall. Mostly, the significant changes in rainfall occur in the evening, sometimes reaching into the early hours of the morning.

For both regions, we find that LH decreases for deforested areas but mostly limited to the afternoon between 12:00-18:00 UTC. Although this reduced LH would result in less moisture availability during the afternoon, it did not affect rainfall amounts significantly, potentially because rainfall frequency peaks between 15:00 UTC and the early morning hours (see Fig 7b), when oceanic moisture may be advected inland from the coast due to increased onshore winds over night. We assess diurnal cycles of $\Delta\theta$, specific humidity changes, 1.5m temperature changes, pressure changes and changes in 10m convergence caused by the changes in pressure (see section 3.3) and 10m winds (see section 3.4) to show that to understand rainfall changes it is crucial to analyze how deforestation affects the dynamics and thermodynamics.

### 3.6.1 Inland Deforestation with Soil Moisture Stress

In this region (Guinea East, 10-8° W), 1950s rainfall totals are highest for the northern half of the 1950s forest (8-10° N) during the afternoon and evening (Fig 8a). 1950s convergence (Fig 8c) is greatest at the coast and positive in locations where the gradient in roughness length (Fig 8b) becomes positive (forest/grass boundaries, at approx. 7° N, 8.2° N and 9° N). The effect of the sea breeze front, visible as a region of positive convergence moving inland over the evening from 6- approx. 7° N, is also apparent in Fig 8c. The regions of positive convergence coincide with the high rainfall patterns. This is a region with extensive deforestation 300-400 km inland (deforestation is patchy from 5.5-7.5° N and more extensive from 8-10.5° N) (Fig 8e and h) where the daytime temperature (Fig 8g) and turbulent flux changes are large. There is a significant increase in rain from 16:00 UTC to the early morning hours and a significant decrease from 10:00-13:00 UTC at 8-10° N (Fig 8d). There is also a general indication of slightly reduced rainfall south of 8° N. Although the convergence field is noisy, positive rainfall changes at 8-10° N tend to occur where convergence increased after deforestation (Fig 8f). Conversely, negative rainfall changes occur where convergence (south of 8° N) or specific humidity (e.g., 8-11° N during the day) decreased. Increased convergence (Fig 8f) is aligned with areas of decreased roughness length (Fig 8e and h) and surface pressure (Fig 8g). There is no impact on the sea breeze front because there is very little deforestation and no significant temperature change near the coast. From 7-8° N on the upwind edge of the main deforestation, there are less convergent conditions over much of the day due to roughness length changes to the north. From 10:00-18:00 UTC, lower pressure to the north further reduces convergence over that area. From 9-11° N we would expect more convergent conditions throughout the day due to roughness length changes. Pressure reductions also have an effect north of 8° N from 7:00-23:00 UTC and particularly from 9-11° N between 16:00-20:00 UTC due to the large temperature changes. North of 8° N there are mostly more convergent conditions throughout the whole day. Although LH and therefore low-level specific humidity decrease from 8-11° N during the day and are partly responsible for reduced rainfall at this time, the specific humidity increases around 7-10° N in the evening (Fig 8i). The drier air is advected north through the evening with moist air being drawn in from the coast, also resulting in increases in Δθ around 9° N after 20:00 UTC (Fig 8i) (see also supplementary Fig S6).

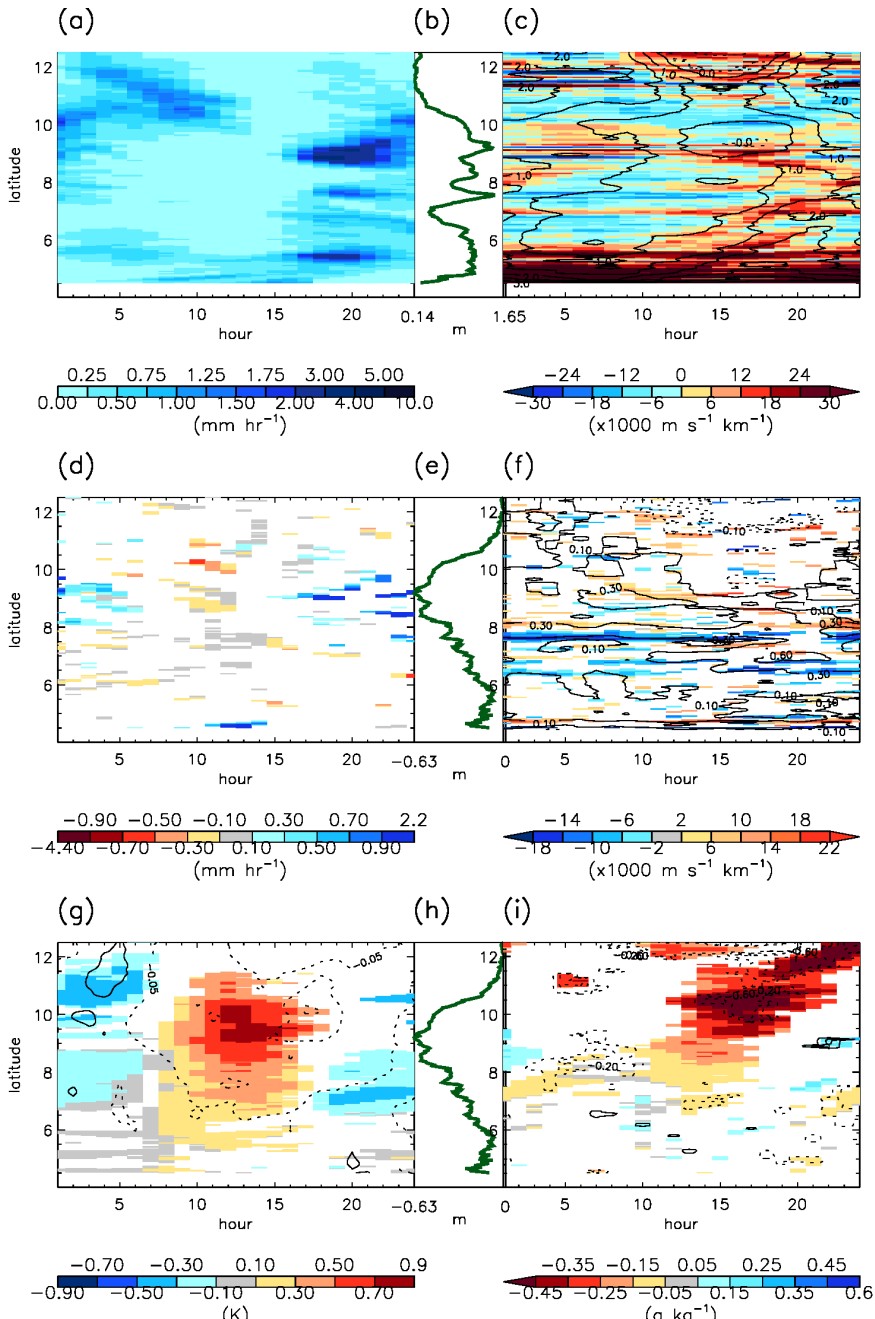

Figure 8: North-south cross sections averaged over 10-8° W over land. Diurnal cycles of (a) 1950s rain rate, (b) 1950s roughness length indicating 1950s forest, (c) 1950s convergence of 10m winds (shading) and 1950s 10m meridional wind (black contours every 0.5 ms$^{-1}$), changes in (d) rain rate, (e) roughness length indicating the extent of deforestation, (f) convergence of 10m winds (shading) with 10m meridional wind (black contours every 0.2 ms$^{-1}$), (g) 1.5m temperature (shading) and surface pressure (black contours every 0.1 hPa), (h) roughness length (repeat of (e)), and (i) specific humidity (shading) with Δθ (black contours every 0.4K). Insignificant changes are shown in white, whereas small but significant changes either side of zero are shown in grey.

Focusing on the area of strongest deforestation between 8-10° N, we find small increases in the number of afternoon initiations in this region compared to the 1950s and increases in the number of storms of 15% throughout the afternoon and evening, which increases to approx. 40% for the early morning hours (Fig 9a and c). An increase in local afternoon storm frequency after deforestation is consistent with observational results for the region (Taylor et al 2022). During the evening, the mean storm area increases by approx. 20% and into the early hours of the morning before 5:00 UTC, mean storm area and intensity increase by approx. 30% with increases in area preceding increases in intensity (Fig 9b and c). These all result in higher rainfall. The lower rainfall seen between 8:00-12:00 UTC is due partly to a smaller number of storms (approx. 25% lower between 8:00-13:00 UTC) but mainly due to the mean storm area being considerably smaller (30-70% between 5:00-14:00 UTC). Note a similar balance of processes is responsible for the enhanced precipitation over a deforested area further west in Guinea (Fig 7c).

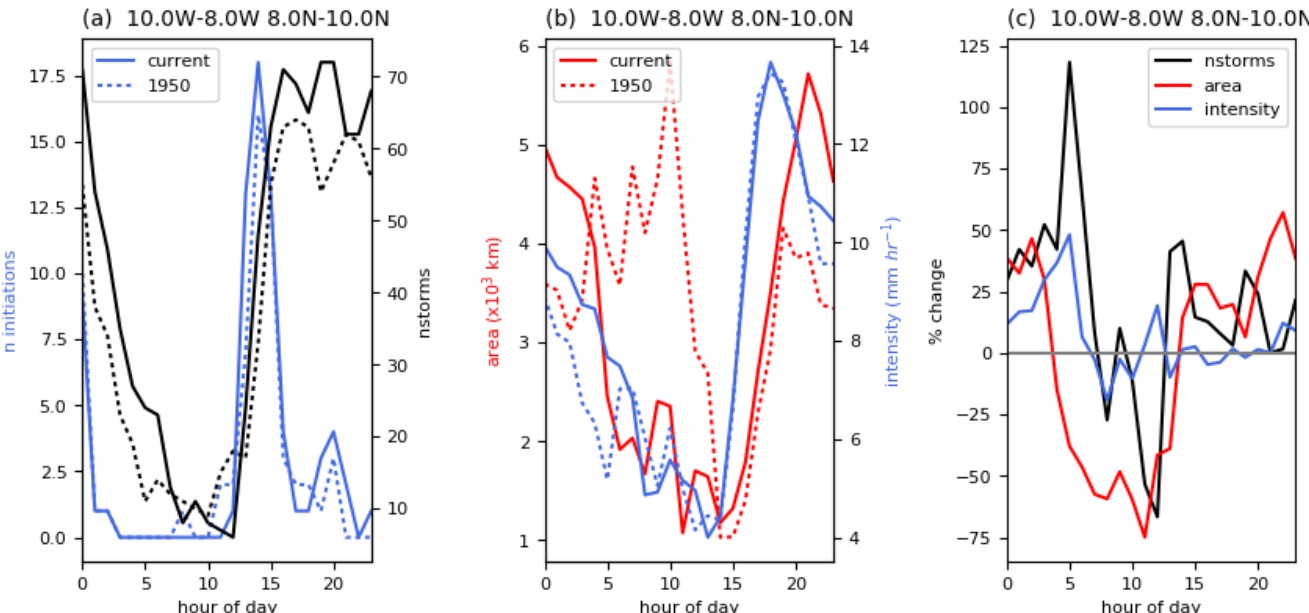

**Figure 9: Diurnal cycles in the region 10-8° W, 8-10° N of (a) number of spontaneous initiations (light blue) and number of storms present (black), (b) mean area (red) and mean intensity (light blue) of storms and (c) the % change in number of storms (black), mean area (red), and mean intensity (light blue).**

### 3.6.2 Coastal deforestation

In this region (Cote d'Ivoire, 6-3° W), 1950s rainfall (Fig 10a) is greatest near the coast over much of the day and driven by
the sea breeze. The strong 1950s onshore winds and the sea breeze convergence are dominant features (Fig 10c) causing the
coastal rain to move inland to 7° N during the evening.

This is an area of near-coastal deforestation (Fig 10e) where the turbulent flux and daytime temperature changes (Fig 10g) are
smaller than in the previous region. This region shows two distinct sub-regions regarding rainfall responses to deforestation
(Fig 10d), caused by a northward shift in rainfall. The rainfall changes follow the changes in convergence (Fig 10f), i.e.,
stronger convergence results in more rainfall. There are only small changes in surface pressure (Fig 10g), showing that the
changes in convergence (mesoscale circulation changes) are mostly due to roughness length changes, i.e., a dynamical rather
than thermal response. Latent heat decreases mostly between 13:00-18:00 UTC from 6-7° N (not shown), resulting in a
decrease in low-level (925 hPa) specific humidity (Fig 10i), yet this has no evident effect on coincident rainfall. See also
supplementary Fig S7 for specific humidity and wind profiles.

On the other hand, from 5-6.5° N there has been a decrease in rainfall between 14:00-0:00 UTC, coincident with a decrease in
convergence over the southern part of the deforestation. In this area, we accordingly find a reduction in the number of
spontaneous initiations in the early afternoon and a reduction in the number of storms (approx. 20%) throughout the day (Fig
11a and c) as well as a reduction in the mean area (approx. 35%) and intensity (approx. 15%) of storms in the evening (Fig
11b and c). The increase in mean area and intensity of approx. 15% between 8:00-13:00 UTC does not result in increased rain
due to the simultaneous reduction in the number of storms.

Looking further north at 6.5-8.0° N, we find an increase in rainfall between 15:00-0:00 UTC in the area of the northern edge
of deforestation linked to more convergent conditions (Fig 10f). Low-level specific humidity around 7° N increases from
approx. 18:00 UTC, with the band of wetter air moving northward during the evening until it reaches 8° N at 0:00 UTC (Fig
10i), trailed by drier more stable (lower $\Delta\theta$) air. We find more initiations from 12:00-15:00 UTC and more storms (approx.
80% increase) from 13:00-0:00 UTC (Fig 11d and f) with larger mean area (approx. 60% increase) from 17:00-0:00 UTC and
an increase in mean rainfall intensity (approx. 20%) from 17:00-22:00 UTC (Fig 11e and f), explaining the increase in total
rainfall.

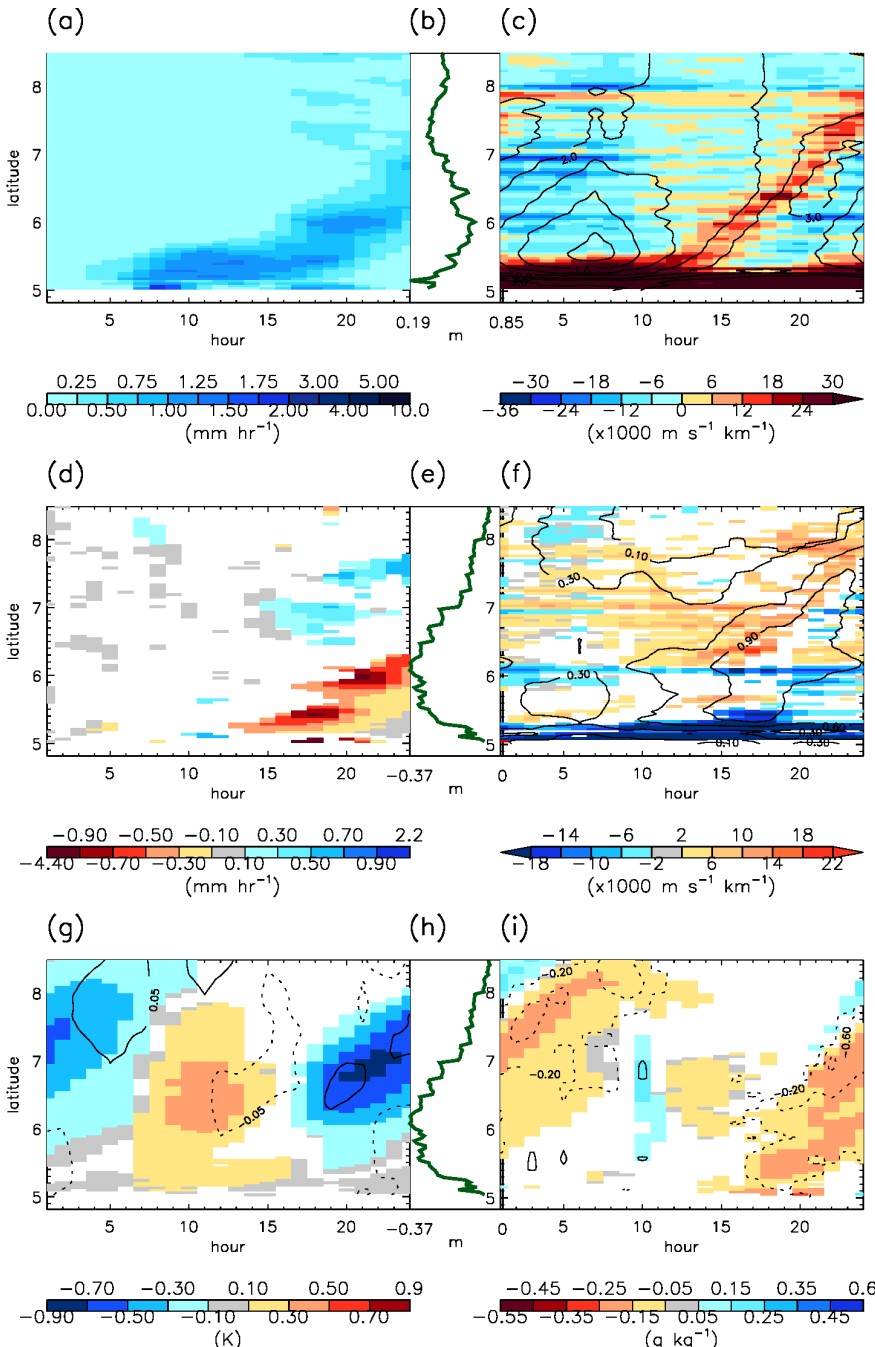

**Figure 10: As for Fig 8 but for cross sections 6-3° W.**

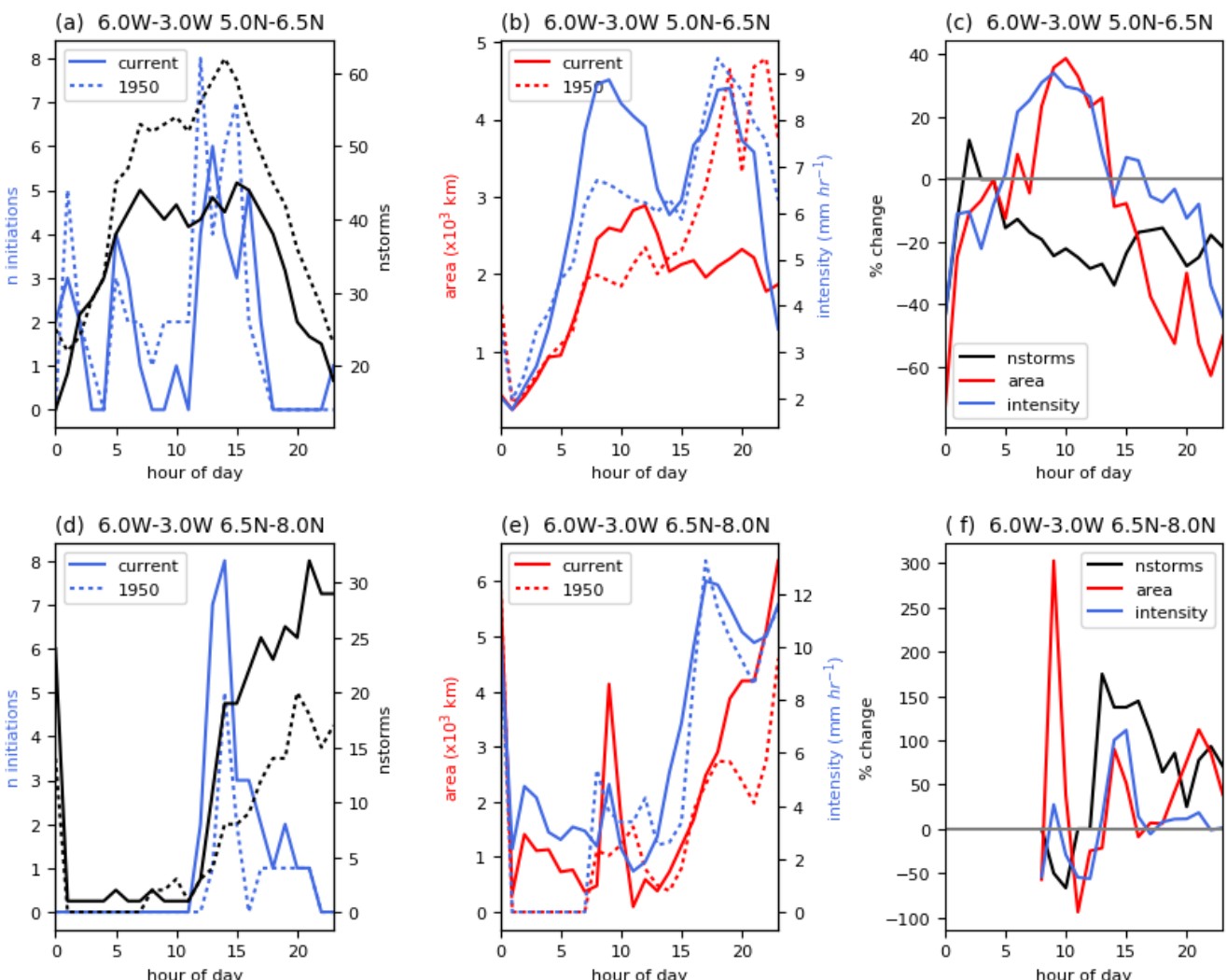

**Figure 11: As for Fig 9 but for cross sections 6-3° W and 2 different sub-regions.**

In this region, where thermal effects of deforestation are minor, changes in convergence caused by reduced surface roughness
drive the sea breeze further inland in the afternoon and evening (see Parker et al. 2017, their Fig 4.18), suppressing convection
in the subsiding air behind the sea breeze front and resulting in a shift in rainfall from 5-6.5° N to 6.5-8° N.

## 4 Comparison with other studies

We now compare our daily mean results over land in the region 16° W-16° E, 0-15° N to three studies which look at the
summer season in West Africa with deforestation scenarios of increasing realism, described in Table 1 – Abiodun et al. (2008)
(A08), Boone et al. (2016) (B16) and Chilukoti and Xue (2020) (C20). It should be noted that our simulations only ran for 5

days in June whereas these studies look at multi-year seasonal changes and therefore are not directly comparable. However, a comparison considering the sign of the changes is justifiable. We would expect the magnitude of our changes to be less than that seen in other studies due to the less extreme deforestation. Our results suggest that much of the deforested area now has enhanced rainfall, with an average increase of 6% over the whole day over deforested pixels. A08, found decreases over the deforested area of 45% and B16 and C20 found decreases in rainfall over all land of 4-25%, although in B16, half the models showed the rainfall shifting south.

Our imposed deforestation scenario (approx. 11% reduction in trees) translates into local albedo increases of up to 0.02 with a mean of 0.0014, and a mean roughness length decrease of 0.38m. The increases in albedo for some models in B16 are an order of magnitude larger than our albedo increases, although some models had similar albedo changes; in A08 the albedo increased by approx. 0.1 (45%); in C20 the albedo increase is similar to ours. In A08, roughness length decreased by between 0.25m and 1.95m (based on disturbed forest or tropical forest changing to short grass, see their table 1). The reduced roughness over grass compared to trees has a significant effect on the 10m winds over deforested areas. As generally over West Africa the 10m winds are south-westerly, this means both 10m U and V increase. Over the whole day we find V increased by 0.23 ms$^{-1}$, a similar magnitude to that seen in A08.

We find the net downward LW radiative flux decreases during the day by 3 Wm$^{-2}$ over the deforested area (due to enhanced surface emission) and net downward SW radiative flux changes were more variable but mostly positive. Our whole day mean values (LW = -1.1 Wm$^{-2}$, SW = +0.3 Wm$^{-2}$) are the same sign as, albeit an order of magnitude smaller than, A08 and C20. Compared to our simulations, the net downward radiative flux decrease was much greater in some models in B16 due to albedo changes being more dominant in those models than reductions in cloud cover.

We find latent heat flux is generally reduced during the day by 4 Wm$^{-2}$ over deforested areas as is expected, with whole day value of -2.3 Wm$^{-2}$. We find sensible heat increases during the day by 5 Wm$^{-2}$ with whole day value of +1.3 Wm$^{-2}$. These are the same sign albeit somewhat smaller than in A08 and C20 and falls within the large range from different models in B16. Deforested areas are mostly warmer during the day (up to 1 K, deforested mean 0.2K) and cooler during the night (down to -0.5 K, deforested mean -0.2K). Effects are constrained to the lower troposphere (lower than 700 hPa). Temperature changes are usually larger when the extent of the deforestation is larger such as in the west of this region of study. Over the whole day, we found no significant change whereas A08, B16 and C20 found up to 1K warming.

Our deforestation scenario is not as extreme in extent or intensity as many previously assessed deforestation scenarios (e.g., A08, B16, C20) and therefore we might not expect the large reductions in rainfall seen previously. A more extreme deforestation scenario may well have greater effect on atmospheric moisture. However, the changes in a number of atmospheric diagnostics are of the same sign and not dissimilar in magnitude to previous studies, yet in contrast, we find increases in rainfall.

|  | Our study (June) | A08 (JAS) | B16 (JAS) | C20 (JJA) |
|---|---|---|---|---|
| Deforestation extent | Reduction in trees of 11% in region | 100% removal of trees south of 17N | Reduction in trees approx. 20% in region | Time dependent reduction in trees up to approx. 30% south of 15° N by 2010 |
| Rain (all land) | +2% | N/A | -4% to -25% | -10% |
| Rain (deforested land) | +6% | -45% | N/A | N/A |
| Albedo | +0.0014 | +0.1 | 0 to +0.14 | <0.01 |
| Roughness length (m) | -0.38 | -0.25 to -1.95 | N/A | N/A |
| LW ($Wm^{-2}$) | -1.1 | -10 | N/A | -8 |
| SW ($Wm^{-2}$) | +0.3 | +5 | N/A | +4 |
| LW+SW ($Wm^{-2}$) | -0.8 | -5 | 0 to -30 | -4 |
| LH ($Wm^{-2}$) | -2.3 | -40 | 0 to -50 | -6 |
| H ($Wm^{-2}$) | 1.3 | +20 | 0 to +40 | +8 |
| T 1.5m (K) | 0 | +1 | 0 to +1 | +0.8 |
| V 10m (m s$^{-1}$) | 0.23 | 0.5 | N/A | N/A |

**Table 1: Comparison of whole day mean variables over 16W-16E, 0N-15N (deforested land unless specified otherwise), estimated from figures in A08, B16 and C20 if not given as region means.**

## 5 Conclusions

We have assessed the impact of recent deforestation in West Africa on rainfall in early June using an ensemble of 5-day simulations with a CPM. Our deforestation scenario is more realistic and less extreme than previous studies and we assessed the changes over the diurnal cycle. We also assessed the statistics of MCSs. Whilst our simulations examine only a brief period within the full annual cycle, the analysis draws out key processes driving the rainfall response. Our results show much of the deforested area now has enhanced rainfall (mostly due to more storms but also bigger storms and to a lesser degree more intense storms) between 18:00-6:00 UTC, with an average increase of 8.4% across all deforested pixels, and whole day increases of 2% (6%) over all land (deforested land) up to 15° N, unlike previous studies. The changes are more significant during the night than the day and how the rainfall changes is strongly dependent on both soil moisture status and the proximity to the coast of the deforestation due to sea breeze interactions. We also find that changes are quite localized, and we did not see marked significant changes in any variable north of 15° N. The African Easterly Jet also did not change position in our model, although this has been found to occur in other studies (B16). However, our simulations only cover a short time period. Had we performed longer simulations allowing large-scale circulation changes to occur, we may have seen shifts in rainfall

on top of the changes presented here. However, it is unlikely that the overall rainfall change would have reversed sign given

the relatively small extent of deforestation in our study compared to those studies showing large-scale circulation shifts and the fact that tropical deforestation has a much smaller effect on the ITCZ than deforestation at higher latitudes (e.g. Devaraju et al., 2015). Our simulations were run with a climatological soil moisture consistent with the vegetation in order to reduce transient changes due to soil moisture not matching the evaporative properties of the underlying vegetation. It is not possible to extrapolate our results to other months as we have shown thermal responses are dependent on soil wetness and location of

the 1950s rain, which differ through the seasons. However, the different mechanisms presented here would still apply albeit producing a different pattern of rainfall change. Future studies in different months and for longer periods of time would thus be beneficial.

For process evaluation, we considered two contrasting deforestation regions in detail. In the first region, where the deforested zone is well inland, we found that soil moisture limitation ensures enhanced sensible heat, which creates thermally-driven

convergence over the deforested zone. Roughness length-induced reductions in convergence to the south and increases in convergence in the north of the deforested zone also occur and rainfall increases locally within the deforested zone. However, in the second region, where wetter soils mean there is little impact of deforestation on boundary layer temperatures, reduced roughness lengths are responsible for changes in rainfall: The deforestation maximum lies approx. 100km from the coastline, and weaker surface drag allows the sea breeze to penetrate faster and further into the interior. The associated changes in

convergence and advection of moist oceanic air effectively reduce rainfall near the coast whilst enhancing it inland. Our results show that deforestation near the coast, where the sea breeze impacts convection, results in very different patterns of rainfall changes than deforestation further inland. A similar conclusion was drawn in the observational study by Taylor et al. (2022). Focusing on deforestation within a coastal strip of 50km, they found enhanced convective activity maximized within that strip, embedded within the sea breeze. The two analyses differ both in the proximity of the deforestation to the coastline, and the

likely role of thermal effects. Considering the location in the current study, the deforestation is further inland, which draws sea breeze convection away from the coastal strip. Secondly, whilst the model analysis focuses on a location and time of year when soil moisture effects are less important, the observational analysis draws from a broader range of conditions in space and time, and thermal effects are expected to be more important.

We have shown that changes in low level winds over deforested areas can cause more convergent conditions into the evening

when MCSs are numerous. Although humidity (and conditional instability) does decrease over deforested areas in the day, it recovers in the evening and can increase in some areas due to enhanced advection inland due to increased winds. However, the changes in convergence in our model were found to be the dominant control on rainfall with little impact from changes in the moisture budget, in line with observed rainfall changes in the region (Taylor et al 2022) and with observed increases in cloud cover over deforested regions in general in the tropics (Xu et al. 2022). Our model adds strong support to the

observationally-based link between deforestation and increased rainfall in the coastal zone (Taylor et al., 2022). Our results illuminate the mechanisms responsible and demonstrate that the processes at work differ from region to region according to the precise geographic environment (i.e. distance from coast) and seasonally (according to overall water stress).

In contrast to PMs, CPMs show enhanced rainfall in areas where there is mesoscale convergence (Birch et al 2014), for example along soil moisture boundaries, in line with observed mechanisms (Taylor et al., 2013), rather than being controlled by the moisture budget. It has also been found that CPMs better capture the rainfall response to sea breeze changes (Finney et al., 2020). PMs simulate the convective peak too early in the day (approx. 13:00 UTC) compared to observations and CPMs. Such models would be more affected by the reduced evaporation caused by deforestation during the early afternoon, as rainfall is strongly controlled by the vertical profile of temperature and moisture, which could explain why previous deforestation studies show a tendency to reductions in rainfall over deforested areas. Therefore, we suggest that the mesoscale convergence produced by deforestation is likely to affect rainfall more than previously found in studies using PMs. It is imperative that further studies using CPMs over extended periods are undertaken to understand whether our findings are representative of all CPMs. We have demonstrated that a CPM can capture the effects of enhanced convergence due to deforestation and in principle could be used to make future projections of changes due to deforestation. However, substantial time was taken initially to improve the land surface model and the surface conditions, and we would not expect such projections yet to be reliable without a comparable effort to ensure that the model setup is good enough (out-of-the-box models have very different representations of land cover and bio-geophysical responses e.g., Pitman et al., 2009; Boone et al., 2016; Boysen et al., 2020). We therefore suggest that both CPM ensembles of short simulations at different times of the year as well as ensembles of longer simulations are performed. But beforehand it is essential to consider:

- How well the land surface model being used represents observed behaviour of the different vegetation types in terms of differences in albedo, leaf area index, ability to access soil moisture and partition surface fluxes accurately under different water stress conditions,

- How to create the vegetation fraction maps to be used for historical deforestation scenarios,

With sustained effort, further improvements could be made to JULES over the simple modifications we made specifically for this study.

**Code/Data availability**

Model output is available at Crook, J. (2021): VERA: West Africa current vegetation and 1950 vegetation scenario ensemble mean data for June 2014. Centre for Environmental Data Analysis, https://catalogue.ceda.ac.uk/uuid/db259fd2bad64b6da9af884121a160a6.

**Author contributions**

Julia Crook performed the model simulations, analyzed the simulation outputs, wrote the manuscript, and curated the data. Cornelia Klein contributed to reviewing and editing the manuscript.

Sonja Folwell developed the 1950s vegetation map, produced the ancillaries required for the simulations, made the modifications required to JULES and contributed to writing the manuscript.

Chris Taylor and Doug Parker acquired funding, performed project administration, and contributed to reviewing and editing the manuscript. Chris Taylor also had significant input into understanding issues with the standard UM model setup.

Adama Bamba and Kouakou Kouadio helped in the analysis of the sea breeze impacts in the Cote d'Ivoire region.

**Competing interests**

The authors declare that they have no conflict of interest.

**Acknowledgments**

This work used the ARCHER UK National Supercomputing Service (http://www.archer.ac.uk) to run the simulations. We thank Willie McGinty (National Centre for Atmospheric Science) for help in setting up the simulations. The work was funded by Natural Environment Research Council (NERC) VERA project (NE/M003574/1) and NERC/DFID AMMA-2050 project (NE/M020126/1). Parker was supported by a Royal Society Wolfson Research Merit Award (2014-2018).

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
