# Peer review of "Vegetation Properties"

_Weather and Climate Dynamics, 2022_

## Author Comment (AC1)

*We thank both anonymous reviews for their time and effort in reviewing our article. We respond to their comments below.*

Reviewer 1

The study pertains to understanding how current west African deforestation impacts the regional hydroclimate as compared to the land cover in the 1950s using a cloud resolving regional atmospheric model. The pre- to early- monsoon period is simulated and various hydro-meteorological variables are analysed to understand the impacts of deforestation of convection, precipitation and underlying processes. The authors show small but significant increases in precipitation due to deforestation unlike the findings of a precipitation decrease by previous studies, which is the novelty of this result and probably also matches with some observations as claimed by the authors. It seems that the use of the cloud resolving simulations have helped the authors to achieve these similar-to-observation results. Overall I think the study and its results are important for publication. However, I do find certain issues with the writing and presentation style which have made the article a difficult read. There are also certain aspects of the study that need clarification in the paper. I suggest a revision of these aspects of the study before final acceptance. I have provided my specific comments below. I have also provided my answers to the questions provided on the journal website -

1. Does the paper address relevant scientific questions within the scope of WCD? Yes

2. Does the paper present novel concepts, ideas, tools, or data? No

3. Are substantial conclusions reached? Yes

4. Are the scientific methods and assumptions valid and clearly outlined? No

5. Are the results sufficient to support the interpretations and conclusions? Yes

6. Is the description of experiments and calculations sufficiently complete and precise to allow their reproduction by fellow scientists (traceability of results)? To some extent

7. Do the authors give proper credit to related work and clearly indicate their own new/original contribution? Yes

8. Does the title clearly reflect the contents of the paper? Yes

9. Does the abstract provide a concise and complete summary? Yes

10. Is the overall presentation well structured and clear? No

11. Is the language fluent and precise? No

12. Are mathematical formulae, symbols, abbreviations, and units correctly defined and used? Almost

13. Should any parts of the paper (text, formulae, figures, tables) be clarified, reduced, combined, or eliminated? Yes

14. Are the number and quality of references appropriate? Yes

15. Is the amount and quality of supplementary material appropriate? Yes

Specific Reviewer Comments –

1. The writing style is confusing and not succinct. At some places important information/explanation seems to be missing. At some times it even sounds more casual than expected for a scientific article. Examples – section 2.1.3, paragraph 1 of section 3.6. Some of the results need to be presented in a better way as well. For example, I found section 3.6 quite difficult to understand in the first reading probably because the results and discussion have not been presented clearly. It is hard to point at more examples, and ways in which improvements can be made, but there is nevertheless a general need for improvement on the writing style. A revision would help make the results more accessible to readers.

   *We have made various changes to the text throughout the document.*

   *We have reworded section 2.1.3 to clarify that for vegetated tiles, a contribution from soil evaporation dependent on the fraction of bare soil visible through the canopy is also included and it is this contribution that is switched off because it has been shown to be too large under some circumstances. We now say "However, for the vegetated tiles, total evapotranspiration is made up of transpiration from the leaves plus a contribution from soil evaporation based on the fraction of bare soil visible through the vegetation canopy. Under some circumstances, this bare soil evaporation in vegetated tiles is known to be too large (Van Den Hoof et al., 2013). We found that this source could (counter-intuitively) enhance evapotranspiration following deforestation and we therefore switched the bare soil contribution off for all plant tiles."*

   *See response to point 4 below regarding section 3.6.*

2. A more detailed discussion of the cloud resolving model is needed. A 4 km spatial resolution falls in the grey area where clouds cannot be simulated explicitly and cannot be parameterized properly because the assumptions of regular cloud parameterizations, based on spatial statistics applicable to larger scales break. So the authors should explain in more detail why they have chosen to work in this intermediate spatial resolution.

   *We accept that 4km is at the edge of the grey zone. However, at this resolution the model has been shown to capture key properties of observed rainfall and storms in West Africa that parameterized models cannot. In section 2.1 we have provided more details about the model and pointed to other studies that show the 4km CP resolution UM does compare favourably with observations:*

   *"The model includes a convection parametrization (Gregory & Rowntree, 1990) with closure based on the convective available potential energy. However, this parameterization is severely restricted by adjusting the relaxation time, and a sub-grid Smagorinsky-type turbulent mixing scheme is employed, allowing explicit convection. While a 4km model is at the edge between the grey zone and truly convection-permitting resolution (Prein et al 2015), this model has been shown to represent the diurnal cycle, the intermittency of convective rainfall, the propagation of convection, the location and the lifetimes of deep convective storms in West Africa more accurately than the equivalent 12km parameterized*

*model when compared to CMORPH rainfall, TRMM radar (2A25) and SEVIRI brightness temperature (Crook et al., 2019). It does, however, have storms that are often too intense and never reach the size of the largest observed storms, and the small storms produce too much of the total rainfall. It has also been shown to capture the observed relationships between surface flux patterns and convective triggering, unlike the 12km parameterized model (Taylor et al. 2013)."*

3. Some evaluation of model results with observations is needed. While there might not be in-situ observations available from this region, there are satellite data products of cloud and precipitation (on larger spatial scales) available which can be used to at least provided a qualitative comparison with observations.

*We agree with the reviewer and have added in section 2.1 the fact that this model does have a good representation of rainfall in the region compared to CMORPH, TRMM radar and SEVIRI (see point 2 above).*

The authors have also mentioned some previous observational studies and that their results relating with precipitation changes agree with these observations. For example on Line 323 authors have referred to Taylor et al. 2021. It would be better if some of these observations are included in this manuscript to (1) validate the baseline simulations and (2) provide comparison to simulated changes.

*We have added references to Crook et al 2019 for validation of baseline simulations in section 2.1 (see point 2 above).*

*We can't directly compare the observed impact of deforestation on rainfall rates with observations because there is no observational estimate of this quantity. However, we can compare aspects of Figure 7b with the analysis of Taylor et al. (2022), who analysed the change in frequency of convective cores over a 30-year period as a function of LST trends (a proxy for deforestation). First we note similarities in the timing of the convective enhancement post-deforestation. The observational analysis showed maximum increases around the late afternoon/early evening convective peak, with enhanced convection persisting downstream for several hours (e.g. their Figure 2). The timing of this enhancement is quite consistent with our simulated result in Figure 7b. We have also done a calculation of the sensitivity of the observed trend in convective activity to deforestation. We took the data presented in Figure 1c of that study and expressed the trend in core frequency relative to its mean frequency (see Figure below). That allows us to compare relative trends in convection observed over the period 1991-2020 with a relative change in simulated rainfall rates over deforested areas (Figure 7b). Changes in both observed convective activity and simulated precipitation are sampled at the diurnal maximum. We assumed an average observed LST trend of 1.5K/decade, consistent with significant deforestation according to Supplementary Figure 3 of Taylor et al. (2022). The linear regression line then yields a relative increase in convective core frequency at the diurnal peak of 18%. This increase is not dissimilar to our relative rainfall increase at 20:00 UTC of*

*9%. We would not expect an exact match given the different nature of the data and that the observations span the entire rainy period in Southern West Africa for a period of 30 years whilst our simulation represents conditions from a single year in early June. We have added text to this effect.*

[Figure]

4. Why have the authors not analysed the changes over regions like Guinea, Sierra Leone, Cameroon and central African Republic where the change in tree cover is the maximum? Although the authors have presented some valid reasons for their choice of analysing the two regions, it is still unclear why they chose these regions over some other very interesting regions in their simulated domain where interesting and larger changes to the hydro-climate have occurred. While there might an obvious reason for this, it is not clear in the writing. Authors should point that out more clearly.

*We apologise that this section was not clear. We did analyse changes over East Guinea (10W-8W) and stated in the results that the behaviour was similar to the region further west at 13W which includes Sierra Leone and West Guinea. We chose not to analyse Cameroon as this is near the edge of the domain and is a mountainous region which likely complicates the response, and the Central Africa Republic is not within our domain and would need to be the subject of a future study. The point of this section is to highlight that different regions have different processes at play depending on proximity to coast, patterns of 1950s rainfall and extent of deforestation. The two regions chosen are representative of that, and we do not believe that analysing every region where rainfall changes are found would make an interesting read or be beneficial. The first part of section 3.6 now reads:*

*"Processes governing rainfall changes are dependent on proximity of the deforestation to the coast, location of 1950s rainfall and strength of the sea breeze, the soil wetness and the extent of deforestation (larger areas of deforestation and drier areas have greater temperature differences). To demonstrate this, we now assess in detail the changes in two specific focus regions (shown in Fig 2g), chosen for their*

*contrasting soil wetness, extent of deforestation and proximity to the coast. In the first case (Guinea East, 10-8° W, Fig 2g box 1), the extensive deforestation is up to 400 km from the coast, a region that is 1-2 months into the rainy season for the simulated period. The recent start of rainfall after the dry season means that evapotranspiration is still limited by soil moisture ~200 km or more inland (i.e., FSMC<1) such that deforestation induces a decrease in evaporative fraction and atmospheric warming. This decrease in evaporative fraction with deforestation is also true for the Sierra Leone/ Guinea West (~13W) region. We thus consider our chosen Guinea East box as representative for the deforested latitudinal band across these regions The second region (Cote d'Ivoire, 6-3°W), with marked deforestation 40-200 km from the south coast, was chosen for its earlier start to the rainy season meaning that soil moisture is not limited and consequently has only a weak control on evaporative fraction during the simulation. Moreover, rainfall in this second case is strongly influenced by the daytime penetration of the sea breeze.  Whether the rainfall changes in these regions are predominantly dynamically or thermally driven depends on the local characteristics, which will we investigate in the following."*

5. Line 205 – it would be helpful to mention here itself which two deforested regions have been analysed.

   *We have added a statement as to which two deforested regions have been analysed, as requested.*

6. Line 285 – there should be more discussion around what the delta theta proxy means. The conclusions from this analysis are also not stated clearly. The usage of this proxy in the following sections is also not effective because the meaning of this proxy and implications of its change are not clearly defined.

   *We have provided a more detailed explanation of the delta theta proxy and added a reference.*

7. Figure 1 and associated text under section 2.2 can go to Supplementary Infromation.

   *This was a novel part of our study and produced a more realistic historical land cover map than previous studies. We consider that this work was necessary to the success of a study like this and is critical to the relevance and reproducibility of our results. We therefore do not wish to move it to supplementary information. We have added a sentence at the beginning of this section stating why this part is so important.*

Reviewer 2

The study applies a 5-day ensemble forecast over West-Africa using a convection permitting regional climate model in order to understand the effects of historical

deforestation within the region. Due to the high spatial and temporal resolution of the climate model data the authors are able to dissect the processes controlling the weather response within West-Africa as a whole and within 2 subregions highlighting how different local conditions can alter the response to deforestation. Overall this is a unique approach within the research on effects of land cover change on climate and it opens a lot of interesting questions worth exploring in future studies. The study is well structured and written clearly and in a comprehensive way as it addresses several variables in order to understand the changes physically. I would recommend the journal to accept this paper after addressing some minor specific questions added here below.

- Research on deforestation in global idealised simulations studies (a.o. Winckler et al., 2017) have tried to separate local (mostly roughness and albedo effects) from remote effects (large scale circulation), as you highlight in line 109 this study mainly focusses on the local effects and the short period of simulation time does not allow (large-scale) circulation aspects to occur and to influence the results. This might be a strong assumption as these large scale effects strongly influence several variables focussed on within the study (e.g. rainfall due to shifts in ITCZ, Devaraju et al.,2015). How important do you think this bias would be for the interpretation of the results? Do you think that the lack of these large scale circulation changes could help explain the differences between your results and the 3 studies compared to in section 4? I feel these aspects although flagged at some points are not fully addressed yet.

*We do not see shifts in the AEJ in our simulations. Our modelling strategy was designed from the start to comprise ensembles of short simulations, from which we could isolate the forest-change-scale responses, without the complication of significant regional changes. Some models in Boone et al 2016 saw a southward shift in the AEJ which would shift the rain further south. If we were to run a simulation for a long time and allow such large-scale circulation changes, we may also see shifts in patterns on top of the local changes discussed in this paper. We show averaged over the whole land area that rainfall increases unlike the Boone et al 2016 models and these increases are triggered by mesoscale convergence which is unlikely to happen in parameterized models. Our deforestation scenario is less extensive than that in Boone et al. (2016) so we would expect smaller circulation changes. Therefore, we do not believe the lack of large-scale circulation changes could explain the differences between our results and theirs. Devaraju et al. (2015) showed that extratropical deforestation produces much larger shifts in the ITCZ than tropical deforestation. Further studies would be needed with convection permitting models run over longer time scales to explore large-scale circulation changes, but this was not the aim of our study. In Section 5 we have added:*

*"Had we performed longer simulations allowing large-scale circulation changes to occur we may have seen shifts in rainfall on top of the changes presented here. However, it is unlikely that the overall rainfall change would have reversed sign given the relatively small extent of deforestation in our study compared to those studies showing large-scale circulation shifts and the fact that tropical deforestation has a much*

*smaller effect on the ITCZ than deforestation at higher latitudes (e.g. Devaraju et al., 2015)."*

- I'm intrigued by the approach of a 5 days forecasting ensemble, as far as I am aware this has not been used to asses effects of land cover changes which adds a strong novelty to the study. However I wonder how generalisable these results are? You highlight the importance of choice of season and month in several locations within the manuscript, but wouldn't some effects have a delay of occurring (e.g. initial wettening due to deforestation but after while drying?). This is a known caveat of the method I presume, but I wonder if this could be overcome by for example running this ensemble longer (eg 30 days)? For clarity I do not request additional simulation, but I think some discussion on these methodological aspects would be interesting to include.

***We agree that the use of short ensembles is a relatively novel approach, and it was planned in our study from the start. Similar approaches have been used by Fletcher et al. 2022 (DOI: [10.1002/qj.4218](10.1002/qj.4218)). We cannot extrapolate our results to different months as we have shown thermal responses are dependent on soil wetness and location of the 1950s rain. It would be interesting to run the same type of simulations but at other times of the year to compare the local effects as well as run longer simulations which would allow large-scale circulation and moisture changes. Our simulations were initialised with soil moisture that was consistent with the vegetation (i.e. taken from long offline JULES simulations). Longer simulations starting with 1950s soil moisture and allowing that to evolve would give the transient changes. However, these would be difficult to interpret without a large ensemble. We have added a discussion regarding this in section 5:***

***"Our simulations were run with a climatological soil moisture consistent with the vegetation in order to reduce transient changes due to soil moisture not matching the evaporative properties of the underlying vegetation. It is not possible to extrapolate our results to other months as we have shown thermal responses are dependent on soil wetness and location of the 1950s rain which differ through the seasons. However, the different mechanisms presented here would still apply albeit likely producing a different pattern of rainfall change. Future studies in different months and for longer periods of time would be beneficial."***

- Due to the unique setup of the study it opens a lot of questions for future research of which you highlight some in section 5. Could you go a bit further and try to give some recommendations for example : How can this study inform future work by earth system models and regional climate models? What would be priorities for development or research based on this work, should models invest in more convection permitting and/ or deeper evaluation and developments of surface scheme? I believe these kind of insights can help guide the model development community greatly. Therefore I would suggest to include something in line of a limitations and outlook section within the manuscript in order to have a general discussion on the implications and weaknesses of this study now some of these aspects are mentioned in the conclusions but I feel you could go further in this discussion.

*We suggest that both CPM ensembles of short simulations at different times of the year as well as ensembles of longer simulations are performed.*

*Before performing any deforestation simulations, it is essential to consider:*

- *How well the land surface model represents observed behaviour of different vegetation types in terms of differences in albedo, leaf area index, ability to access soil moisture and partition surface fluxes accurately under different water stress conditions*
- *How to create the vegetation fraction maps to be used for realistic historical deforestation scenarios.*

*With sustained effort, further improvements could be made to JULES over the simple modifications we made specifically for this study.*

Technical corrections:

line 49-52: There is a useful review by Perugini etal (2017) who also have a similar conclusion

*Thank you for pointing this out. We did use this paper to inform this section. We have added "…but changes to rainfall due to deforestation have largely only been studied in models (Perugini, et al. 2017)."*

line126-127: The LUCID studies by Pitman et al. 2009 indeed show this but more recently also Boysen et al. 2020 showed in the LUMIP deforest_glob runs that there still remains large issues and uncertainty within ESMs.

*Thank you for suggesting this reference. We have added "and even in the most recent earth system models there is a large difference in behaviour (Boysen et al. 2020)."*

Line 260: More recently Duveiller et al 2020 (https://doi.org/10.1016/j.landusepol.2019.104382) have a more comprehensive dataset of near surface temperature using similar approaches as Alkama and Cescatti, 2016

*Thank you for suggesting this reference which we have added.*

Figure 9 and 11: I found it a bit unclear what all the lines were indicating on the plots (I initially overlooked the different colours of the labels) perhaps this is my own fault but to help people like me I would suggest to add the colour of the lines between brackets after the variable is introduced in the subscript (e.g. (a) number of spontaneous initiations (green) and number of storms present (blue)). Additionally I find the colours of the last panels (c in Figure 9 and c and f in Figure 11) very similar between nstorms and intensity, I would suggest to change it to a more different colour.

*We have added the colours in the figure caption. We have changed the colours round so that black, red and light blue are used in third column panels to avoid use of 2 blues in the same plot.*

---

## Referee Report (RR1)

**Effects on Early Monsoon Rainfall in West Africa due to Recent Deforestation in a Convection-permitting Ensemble**

**By Julia Crook et al**

The paper addresses the effects of historical deforestation in West Africa on rainfall and other meteorological variables in the same region. They use a seemingly novel approach based on 5-day convection-permitting ensemble forecasts coupled to a modular land surface model which allows them to attribute regional meteorological changes directly to vegetation-induced changes in surface fluxes, roughness, and albedo without large-scale circulation feedbacks. After a regional scale discussion of the results they focus on two regions with contrasting initial soil moisture conditions and analyze processes related to rainfall in detail.

I was asked to evaluate if the authors responses to Reviewer 1 were sufficient and I was not reviewer in the previous round. In addition, I provided some comments that hopefully help the authors to further improve their manuscript. My main concern is the statistical significance testing (see comment below). The paper is clearly written and structured. My impression is that this is a creative and novel approach and I'd be happy to see it published in WCD once the remaining issues are addressed.

In the following I briefly evaluate the responses separately for each of the reviewers comments. In my impression only for comment 3 it is not clear if the authors addressed the reviewers concerns appropriately.

- 1. I find the section 2.1.3 now clearly written even though I can't judge how reasonable the switching off of soil evaporation is. In general, I find the paper mostly clearly written.
- 2. The model is now well described, the motivation for its use is stated and the limitations are mentioned.
- 3. If I understand the comment of Reviewer 1 correctly, the authors are asked to compare their modeled precipitation to observations. As the starting date of the runs is 1st June 2014, rainfall should be compared to measurements during 1st 5th June. The authors elaborate on why they can't use observations directly but I'm not 100% convinced. If I understand correctly, the "current" forecast is more or less a "real" forecast for 1st 5th June 2014. Comparing this output to observations of that period should be possible and I think important to do in this study. Of course the focus of this study is not on forecast validation but on the effect of deforestation. Nevertheless, a comparison to observations would provide important context.

- 4. The authors well explain their rationale to selected the two study regions
- 5. Ok
- 6. Ok
- 7. Ok

**Comment**

I appreciate that the authors used a statistical significance test. However, there needs to be a clearer description of the method used. Importantly, did the authors account for multiple testing, i.e. was the false discovery rate controlled (Wilks, 2016)? If not, this is an issue because significance could just emerge by chance when so many tests (over the whole study domain) are done. This correction could be done, e.g. with a Benjamini-Hochberg correction (for python, see

https://www.statsmodels.org/dev/generated/statsmodels.stats.multitest.multipletests.htm]). Also this correction could help remove some of the patchiness of some of the plots such that there is more focus on the dominant differences.

**Minor comments**

L19 "we for the first time estimate"  $\rightarrow$  I'm not a native speaker but to my ears it sounds better to say "we estimate for the first time", or maybe remove "for the first time" completely

L25 "thermally induced enhanced"  $\rightarrow$  "thermally enhanced"

L43 Unclear if you only talk about the biogeophysical changes or also the biogeochemical changes. With regard to the former it seems more precise to say "local (surface) warming"

L79: They found that the enhanced (?)

L156: real life  $\rightarrow$  reality

L204: Figure 2, which indicates the simulated region, shows maps of  $... \rightarrow$  Figure 2 shows maps of (...) in the target region

L206ff: I suggest revising this section and potentially split it in two. The first paragraph is about the statistical significance test so it could have its own section named "Statistical significance test". The second paragraph is in principle only about the criterion for the definition of deforestation. This topic already appears in Fig 2g which is referred to in the previous section (section 2.2). Hence, the description of the criterion for deforestation could be simply added in section 2.2. In my opinion, there is no need to describe that you look at two focus regions or that you first compare albedo etc. This could be part of an introductory paragraph to the results section and not the methods section. I'm aware this is also a bit a matter of taste but I just feel it improves readability.

Further, the first paragraph of section 2.3 suggests that for albedo, surface roughness, and initial soil moisture you also use a T-test to assess statistical significance. However,

it is unclear to me how this can be done given that (if I understand the methodology correctly) for these variables you only have one field for 1950 and one for current condition (i.e. no ensemble members). If you don't use a T-test, then there is no need to mention these variables in this section.

L235: highFSMC  $\rightarrow$  high FSMC

L244: Detail but I find LH as abbreviation of latent heat flux more intuitive. This would also be consistent with the abbreviation of sensible heat flux (SH)

L250: strong, radiative  $\rightarrow$  strong, radiative

L262ff: Sentence structure is unclear. Why does net downward long-wave radiative fluxes decrease? And what does this have to do with reduced roughness length? Is it that reduced roughness length warms the near surface (as a result of reduced heat land-atmosphere heat fluxes) which, as near surface temperature rises, leads to larger upward long-wave radiative fluxes? This will then, when downward long-wave remains constant, lead to a decrease in *net* downward long-wave radiative fluxes. If this is how we need to think about it I would appreciate a bit a clearer explanation here.

L268: may dominate. , → remove period

L271: increases long-wave emission: not sure if it is clear that this refers to long-wave emission by the surface, and not by the atmosphere. Maybe a clarification would help.

L360: these regions.  $\rightarrow$  period missing

L375: whenoceanic  $\rightarrow$  when oceanic

L377ff: I find the last part of this sentence not very clear. Maybe try: "to show that to understand rainfall changes it is crucial to analyze how deforestation affects dynamics and thermodynamics"

L382 and elsewhere: To improve readability, I suggest to use approximately or approx. instead of  $\sim$

L384: "The regions of positive convergence coincide with the high rainfall patterns." Not sure if I agree. If I compare Fig. 8a and 8c I see that convergence and rainfall coincide sometimes but not always/everywhere. Do you refer only to a certain part of the plot? If yes, it would be helpful if it was specified which part.

**References**

Wilks, D. S. (2016). "The Stippling Shows Statistically Significant Grid Points": How Research Results are Routinely Overstated and Overinterpreted, and What to Do about It, *Bulletin of the American Meteorological Society*, *97*(12), 2263-2273. Retrieved Jan 12, 2023, from https://journals.ametsoc.org/view/journals/bams/97/12/bams-d-15-00267.1.xml

---

## Author Response (AR2)

**We thank both anonymous reviewers for their time and effort in reviewing our article. We respond to the comments of Reviewer 3 below.**

**Reviewer 3**

In the following I briefly evaluate the responses separately for each of the reviewers comments. In my impression only for comment 3 it is not clear if the authors addressed the reviewers concerns appropriately.

 I find the section 2.1.3 now clearly written even though I can't judge how reasonable the switching off of soil evaporation is. In general, I find the paper mostly clearly written.
The model is now well described, the motivation for its use is stated and the limitations are mentioned.

3. If I understand the comment of Reviewer 1 correctly, the authors are asked to compare their modeled precipitation to observations. As the starting date of the runs is  $1_{st}$  June 2014, rainfall should be compared to measurements during  $1_{st} - 5_{th}$  June. The authors elaborate on why they can't use observations directly but I'm not 100% convinced. If I understand correctly, the "current" forecast is more or less a "real" forecast for  $1_{st} - 5_{th}$  June 2014. Comparing this output to observations of that period should be possible and I think important to do in this study. Of course the focus of this study is not on forecast validation but on the effect of deforestation. Nevertheless, a comparison to observations would provide important context.

In our manuscript we referred to our previous paper (Crook et al. 2019) that compared this model with precipitation observations for the whole of June and July. However, we accept that a direct comparison for just the 1st 5 days of June 2014 can be made, although our simulations are not a forecast, and have now added this comparison in section 2.1 with an extra figure in the supplementary material. The supplementary figure illustrates that the modelled 5-day total rainfall does indeed show biases compared to CMORPH for certain regions, most pronounced along coastlines and topography. However, systematic regional biases that affect both our forested and deforested simulations equally will not affect the rainfall change signal linked to deforestation that we are interested in. Given this is a process study, it is the model skill in correctly capturing rainfall timing within the diurnal cycle and in representing the characteristics of convective storms (as demonstrated in Crook et al 2019, simulation V CP4 therein) that is most important for this work. The realistic representation in timing, storm lifetimes and storm precipitation intensities provides confidence in our model results on convection responses when surface roughness and flux patterns change locally due to deforestation.

- 4. The authors well explain their rationale to selected the two study regions
- 5. Ok
- 6. Ok
- 7. Ok

**Comment**

I appreciate that the authors used a statistical significance test. However, there needs to be a clearer description of the method used. Importantly, did the authors account for multiple testing, i.e. was the false discovery rate controlled (Wilks, 2016)? If not, this is an issue because significance could just emerge by chance when so many tests (over the whole study domain) are done. This correction could be done, e.g. with a Benjamini-Hochberg correction (for python, see

https://www.statsmodels.org/dev/generated/statsmodels.stats.multitest.multipletests.html ). Also this correction could help remove some of the patchiness of some of the plots such that there is more focus on the dominant differences.

We had not previously accounted for this. We would like to point out that our data is high resolution with N=250,478 grid points in the maps in our figures. This imposes a very strict limit on the pvalues and as a result applying this to patchy fields such as rainrate, convergence, delta  $\theta$  and SW, results in not being able to find any sorted pvalues that meet the FDR test < alphaFDR x i/N (where alphaFDR is alpha\*2) and therefore we cannot calculate a new alpha to use to test for significance. If we use a much-reduced region where changes have occurred (e.g. using the grid points where the individual T test pvalue<=alpha) we get all these same grid points being significant when using the FDR corrected test. For several of our variables assessed we would not expect a change over the whole region and therefore determining field significance does not seem relevant. We have, therefore, used the Benjamini-Hochberg FDR correction for maps of all variables that are not patchy (i.e. do change over a large part of the domain) and have left the plots of other variables showing the individual test significance. We have highlighted in the text where this is the case.

**Minor comments**

L19 "we for the first time estimate"  $\rightarrow$  1'm not a native speaker but to my ears it sounds better to say "we estimate for the first time", or maybe remove "for the first time" completely

Corrected.

L25 "thermally induced enhanced"  $\rightarrow$  "thermally enhanced"

Corrected.

L43 Unclear if you only talk about the biogeophysical changes or also the biogeochemical changes. With regard to the former it seems more precise to say "local (surface) warming"

We have added "local (near surface)".

L79: They found that the enhanced (?)

**Corrected.**

L156: real life  $\rightarrow$  reality

Corrected.

L204: Figure 2, which indicates the simulated region, shows maps of  $\dots \rightarrow$  Figure 2 shows maps of (...) in the target region

**Corrected.**

L206ff: I suggest revising this section and potentially split it in two. The first paragraph is about the statistical significance test so it could have its own section named "Statistical significance test". The second paragraph is in principle only about the criterion for the definition of deforestation. This topic already appears in Fig 2g which is referred to in the previous section (section 2.2). Hence, the description of the criterion for deforestation could be simply added in section 2.2. In my opinion, there is no need to describe that you look at two focus regions or that you first compare albedo etc. This could be part of an introductory paragraph to the results section and not the methods section. I'm aware this is also a bit a matter of taste but I just feel it improves readability.

We agree with the reviewer that this section can be modified as suggested. Section 2.3 is now called Statistical Significance Tests and we have added information about the FDR correction procedure. The sentences regarding what results we show have been moved to the start of the Results section and the paragraph about the definition of what is counted as deforestation has been moved the end of the previous section.

Further, the first paragraph of section 2.3 suggests that for albedo, surface roughness, and initial soil moisture you also use a T-test to assess statistical significance. However, it is unclear to me how this can be done given that (if I understand the methodology correctly) for these variables you only have one field for 1950 and one for current condition (i.e. no ensemble members). If you don't use a T-test, then there is no need to mention these variables in this section.

Albedo is calculated as the ratio of outgoing SW/ incoming SW at 13:00 UTC on each day for each ensemble member. Roughness length is output by the model daily and therefore can be determined on each day for each ensemble member. The initial soil moisture stress factor (FSMC) is calculated using the soil moisture on the first day of each ensemble member. Therefore, this did allow us to theoretically determine statistical significance. However, given that we would expect virtually the same albedo and roughness length throughout the simulation, and the same value of initial soil moisture in all ensemble members, the statistical significance is not very meaningful, and we now present these as simple differences with no mention of T tests.

**L235: highFSMC $\rightarrow$ high FSMC**

**Corrected.**

L244: Detail but I find LH as abbreviation of latent heat flux more intuitive. This would also be consistent with the abbreviation of sensible heat flux (SH)

**Changed LE to LH throughout.**

L250: strong, radiative  $\rightarrow$  strong, radiative

**Corrected.**

L262ff: Sentence structure is unclear. Why does net downward long-wave radiative fluxes decrease? And what does this have to do with reduced roughness length? Is it that reduced roughness length warms the near surface (as a result of reduced heat land-atmosphere heat fluxes) which, as near surface temperature rises, leads to larger upward long-wave radiative fluxes? This will then, when downward long-wave remains constant, lead to a decrease in *net* downward long-wave radiative fluxes. If this is how we need to think about it I would appreciate a bit a clearer explanation here.

Yes, you are almost correct. The reduced roughness length reduces turbulent fluxes (land to atmosphere fluxes), warming the surface and increasing LWu which decreases LW with no change in LWd. There is also reduced cloud cover which decreases LWd so that also has some effect on decreasing LW. We have rewritten these sentences.

L268: may dominate. ,  $\rightarrow$  remove period

Corrected.

L271: increases long-wave emission: not sure if it is clear that this refers to long-wave emission by the surface, and not by the atmosphere. Maybe a clarification would help.

We have added "from the surface".

L360: these regions.  $\rightarrow$  period missing

Corrected.

L375: whenoceanic  $\rightarrow$  when oceanic

Corrected.

L377ff: I find the last part of this sentence not very clear. Maybe try: "to show that to understand rainfall changes it is crucial to analyze how deforestation affects dynamics and thermodynamics"

Corrected as suggested.

L382 and elsewhere: To improve readability, I suggest to use approximately or approx. instead of  $\sim$

We have changed all occurrences of "~" to either "approx." or "around" as suggested.

L384: "The regions of positive convergence coincide with the high rainfall patterns." Not sure if I agree. If I compare Fig. 8a and 8c I see that convergence and rainfall coincide

sometimes but not always/everywhere. Do you refer only to a certain part of the plot? If yes, it would be helpful if it was specified which part.

We agree that not all positive rainfall and convergence changes coincide. We have modified this to say "Although the convergence field is noisy, positive rainfall changes at 8-10° N tend to occur where convergence increased after deforestation."